# Grounded ridge detection and characterization along the Alaska Arctic coastline using ICESat-2 surface height retrievals

Kennedy A. Lange[1], Alice C. Bradley[1], Kyle Duncan[2], and Sinéad L. Farrell[3]

[1]Geoscience Department, Williams College, 18 Hoxsey St. Williamstown, MA, USA
[2]Earth System Science Interdisciplinary Center (ESSIC), University of Maryland College Park, College Park, USA
[3]Department of Geographical Sciences, University of Maryland College Park, College Park, USA

**Correspondence:** Alice Bradley (alice.c.bradley@williams.edu)

**Abstract.** Grounded sea ice ridges are important morphological features that stabilize shorefast ice along Arctic coastlines. Investigating first the development of shorefast ice around Utqiaġvik, AK, we employ high-resolution altimetry data from NASA's ICESat-2 satellite to identify grounded ridges and to track the development of shorefast ice over the winter season. We apply the University of Maryland Ridge Detection Algorithm (Duncan and Farrell, 2022) using ICESat-2 ATL03 elevation data to identify and calculate ridge sail heights and estimate ridge depths using empirical relationships based on first-year ice ridge geometries surveyed in the Beaufort, Chukchi, and Bering Seas. The estimated ridge depths are then compared with 15 arc-second resolution bathymetric data from the General Bathymetric Chart of the Oceans (GEBCO) to detect likely grounded ridges. This approach for identifying and characterizing grounded ridges in shorefast ice is then applied across 1,500 km of the Alaska Arctic coastline in the 2021-2022 winter to characterize 2442 grounded ridge depth, height, grounding width, and distance from shore. Using a range of depth estimates, each ridge is classified as high, medium, or low confidence. We find that grounded ridges along the Chukchi Sea coast tend to be located closer to shore, wider, and ground at shallower depths than those along the Beaufort coast. High confidence ridge detections tend to be taller, shallower, and closer to shore than lower confidence retrievals along both coastlines. Only 12% of all ridge detections in the Chukchi Sea and 28% in the Beaufort Sea are located within the traditional "stamuki" zone ($\geq 10$ m). Finally, seasonal signatures can be identified in the data despite the low temporal resolution product, suggesting that subsequent events not only increase the height of near-shore ridges but also form additional ridges at greater bathymetric depths throughout the season. With further application of the methods demonstrated here, we can begin to map patterns in shorefast ice stability, seasonality, and improve our understanding of near-shore ice dynamics across Arctic coastal regions in a changing climate.

## 1 Introduction

Coastal sea ice serves as an important barrier between the shoreline and drifting pack ice. Stable landfast ice not only shields the coastline from erosion caused by winter storms (Hošeková et al., 2021; Lantuit and Pollard, 2008; Overeem et al., 2011; Zhang et al., 2004), but also provides a stable surface for transportation between towns, to and from the flaw leads throughout the winter (Baztan et al., 2017), and out to off-shore drill rigs (Potter et al., 1981; Masterson, 2009). Indigenous communities

and mammals rely on this stable ice for food security and sustenance from hunting migratory marine species (Huntington et al., 2022; Laidre et al., 2008; Laidler et al., 2009), birds (Lovvorn et al., 2018), and bowhead whale (Druckenmiller et al., 2010).

An increase in ice breakout events and shorter shorefast ice seasons during recent years endangers hunters whose safety depends on knowledge of ice dynamics (Gearheard et al., 2006; Mahoney et al., 2014; George et al., 2004). Since 1980, the volume of ice in the Arctic Ocean is thought to have declined by 75% (Overland et al., 2014), and observations show that later ice freeze-ups and earlier thaws are occurring throughout the Arctic as a result of shortening winters (Mahoney et al., 2014; Stammerjohn et al., 2012) and thinning ice (Kwok et al., 2009; Rothrock et al., 1999; Laxon et al., 2013; Mahoney et al., 2009; Howell et al., 2016; Gerland et al., 2008). Mahoney et al. (2014) show that dates of first ice, break-up, and ice-free conditions for landfast ice in the Beaufort and Chukchi Seas are changing up to 1 week/decade with later formation and earlier breakup dates each year, excluding break-up dates in the Beaufort which shows no conclusive trend.

Grounded ridges play a crucial role in stabilizing shorefast ice, so better information about the state of these features at both local and pan-Arctic scales can inform decision making around the use of the coastal zone (Eicken et al., 2009). Grounded ridges form either when a compression ridge drifts into shallower waters and gets stuck in the sea floor as water depths decrease or from an in situ collision between the shorefast ice and drifting pack ice (Mahoney et al., 2007b; Jones et al., 2016; Reimnitz et al., 1978), illustrated in 1. Shore ice forms in the early freeze-up season, frozen to the sea floor on or near the beach (1A). As a result of winds or currents, deep keeled ridges (1B) in the pack ice may drift in towards shore and get stuck in the seafloor as water depth decreases. Alternatively, drifting pack ice may collide at the right angle with the shore ice for it to not break off, but rather form a new deformation ridge at the boundary (1C). Ridges that are deep enough to anchor the landfast ice in the sea floor are termed grounded ridges, which stabilize nearshore sea ice along all Arctic coastlines (1D).

The stabilizing mechanism in landfast ice, illustrated in the cross-sectional view on Figure 1, relies on attachment points of the ice both from bottomfast ice along the shoreline and the grounded ridge keel. These stable ridges protect the shorefast ice from drifting pack ice, keeping it immobile until breakup occurs in the spring (Jones et al., 2016). The rates and mechanisms of shorefast ice breakup are determined by the presence and location of grounded ridges, such that firmly grounded ice takes longer to break up (Petrich et al., 2012).

Where a ridge grounds relative to shore depends on several factors: the local bathymetry (water must be shallow enough for the keel to reach the sea floor (Selyuzhenok et al., 2015)), the thickness of the level ice (seasonal timing of ridge formation (Barker and Timco, 2017)), the direction and momentum of drifting pack ice causing the collision event, and the chance location of weak points in the shore ice allowing for brittle failure and ridge formation (Jones et al., 2016). There are both geographic patterns (e.g., bathymetry, aspect of shoreline relative to dominate ice drift direction) and stochastic processes (e.g., weather, timing of freeze-up, and rate of ice growth) influencing the location of grounded ridges in any given winter season. Assuming some typical level sea ice thickness and pack ice momentum, there is a bathymetric depth, called the "stamuki" zone, where the probability of a ridge becoming grounded becomes reasonably high. Over the years, it has been defined in multiple ways: between 15-50 m depths (Barnes et al., 1987), around the 20 m isobath (Barnes et al., 1982), 10-27 m water based on field observations in 1983 (McGonigal and Barrette, 2018). In our analysis, we consider the "stamuki" zone as grounding depths greater than 10 m (Reimnitz et al., 1978).

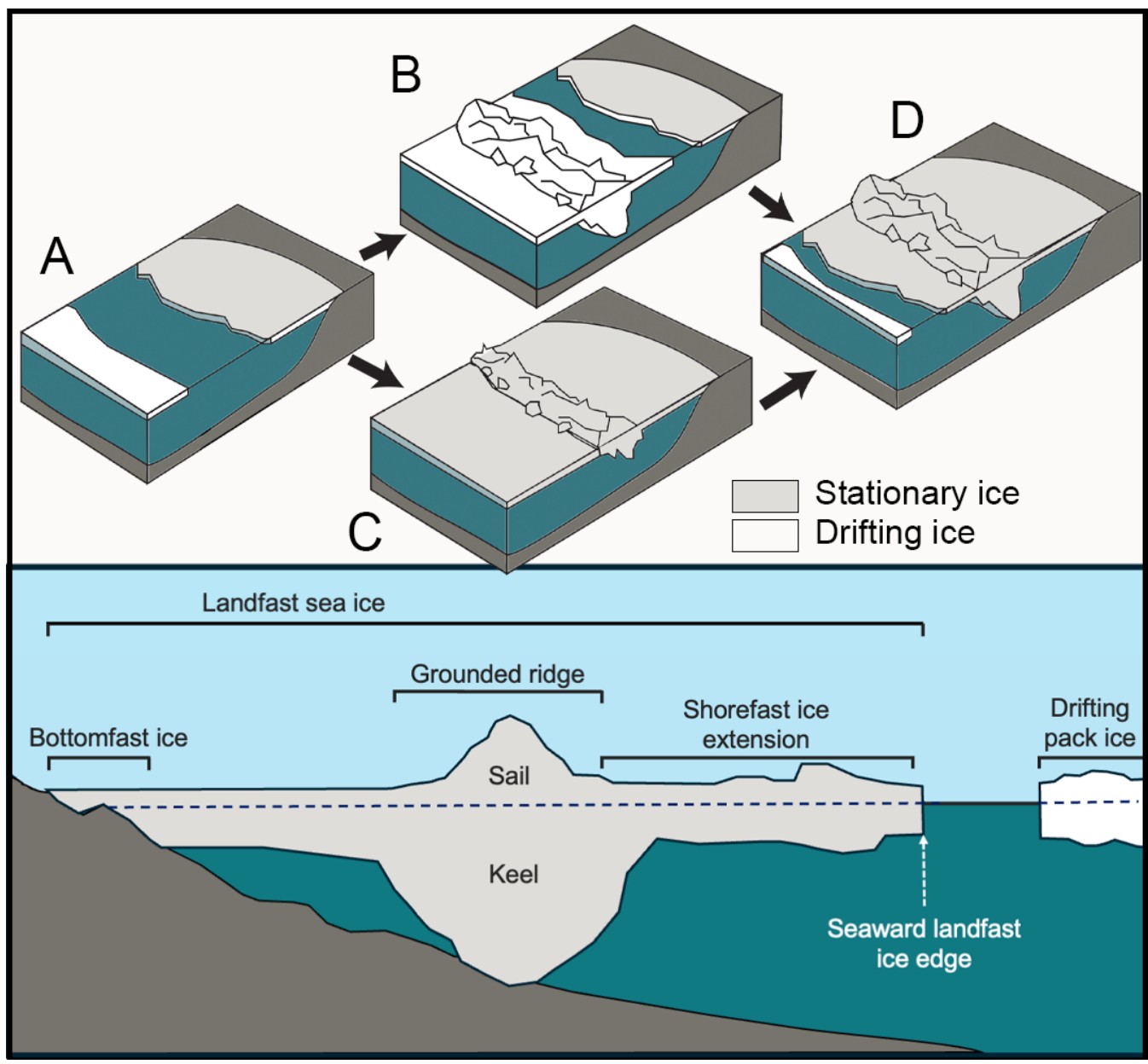

**Figure 1.** Two mechanisms drive the grounded ridge formation process with (A) the formation of bottom-fast ice attached to shore, (B) a ridge drifts into shallow water and becomes grounded or (C) drifting pack ice colliding to form a deformation ridge, and culminating in (D) the ridge is deep enough to embed in the sea floor. The lower panel shows a cross section of a grounded ice ridge during the winter, not to scale. I is bottomfast ice, II is grounded shorefast ice, III is the grounded ridge, IV is the landfast ice extension, and V is drifting pack ice. The seaward landfast ice edge (SLIE) represents the boundary between shorefast ice and a flaw lead or drifting pack ice. Throughout the diagram, gray shading indicates stationary ice.

The location of the landfast ice edge can be identified in satellite products including synthetic aperture radar (SAR, e.g.,

Meyer et al. (2011); Mahoney et al. (2014)), imagery including MODIS (Moderate Resolution Imaging Spectroradiometer e.g., Cooley and Ryan (2024)), and Landsat (e.g. Barry et al. (1979)), and field methods such as electromagnetic ice thickness surveys (e.g. Selyuzhenok et al. (2017)). Studies and observations have documented the extent of landfast ice (Dammann et al., 2019; Selyuzhenok et al., 2015), including significant reductions in landfast ice thickness (George et al., 2004; Mahoney et al., 2009; Gerland et al., 2008) and a shortening stable ice season (Cooley et al., 2020; Meyer et al., 2011; Jensen et al., 2020;

Mahoney et al., 2007a). Cooley et al. (2020) suggests that under a modeled high emission scenario (RCP8.5), the shore-fast ice season will see between a 5 and 44 day reduction in length by 2100. Other studies have found similar results where shore-fast ice seasons are more variable and shortening in duration by around -7 ($\pm$1.5)% per decade (Yu et al., 2014) or by 2.8 days per year (Selyuzhenok et al., 2015).

Determining whether a particular ridge is grounded has previously required in situ manual measurement or under-ice sensors,

and many such surveys predate modern open data policies. The Utqiaġvik sea ice radar system has enabled observations of ridge formation (Mahoney et al., 2007b), though this approach still does not necessarily confirm whether ridges are grounded, and relies upon substantial infrastructure in the form of the sea ice radar system that cannot be applied broadly in more remote regions. Past studies have used remote sensing data to document the stability of ice by measuring movement over time, since ice that remains motionless for extended periods is likely to include grounded ridges as anchoring points (Jones et al.,

2016; Meyer et al., 2011). Even then, remote sensing of the Arctic region is limited by persistent cloud cover, polar night, low-resolution products incapable of resolving small scale ice features, and few satellites that operate on polar orbits. These limitations underscore the importance of developing a method to identify and monitor grounded ridges, as their importance to Arctic communities surpasses the limited attention they have received in the literature.

In this paper, we present a novel method for identifying grounded ridges in shorefast ice over the course of a winter season.

We detect individual ridges in ICESat-2 laser altimetry profiles of surface height using the University of Maryland Ridge Detection Algorithm (UMD-RDA) (Duncan and Farrell, 2022), and estimate their keel depths using empirical relationships of ridge geometries from observed first-year ice ridges in the region. We show that this approach can be used to identify grounded ridges that either persist over time, or are newly grounded, and correspond to features identified in Sentinel-1 SAR imagery. This method was developed using a pilot study area outside Utqiaġvik, Alaska and then applied across the Alaska Arctic

coastline to survey grounded ridges during 2021-2022 winter season. From this analysis, we characterize Arctic grounded ridges and enhance our understanding of Arctic shorefast ice dynamics.

## 2 Data

We used the coastline outside Utqiaġvik Alaska as a pilot area for methodology development and testing, with each side of Point Barrow featuring a unique bathymetric profile and different characteristic ice drift direction relative to shore (Figure 2).

After demonstrating the approach, we extend the analysis along the Alaska Arctic coastline from Point Hope, AK to Komakuk Beach, Canada. While this represents only a small portion of the Arctic, through the analysis of 1271 ICESat-2 ground tracks

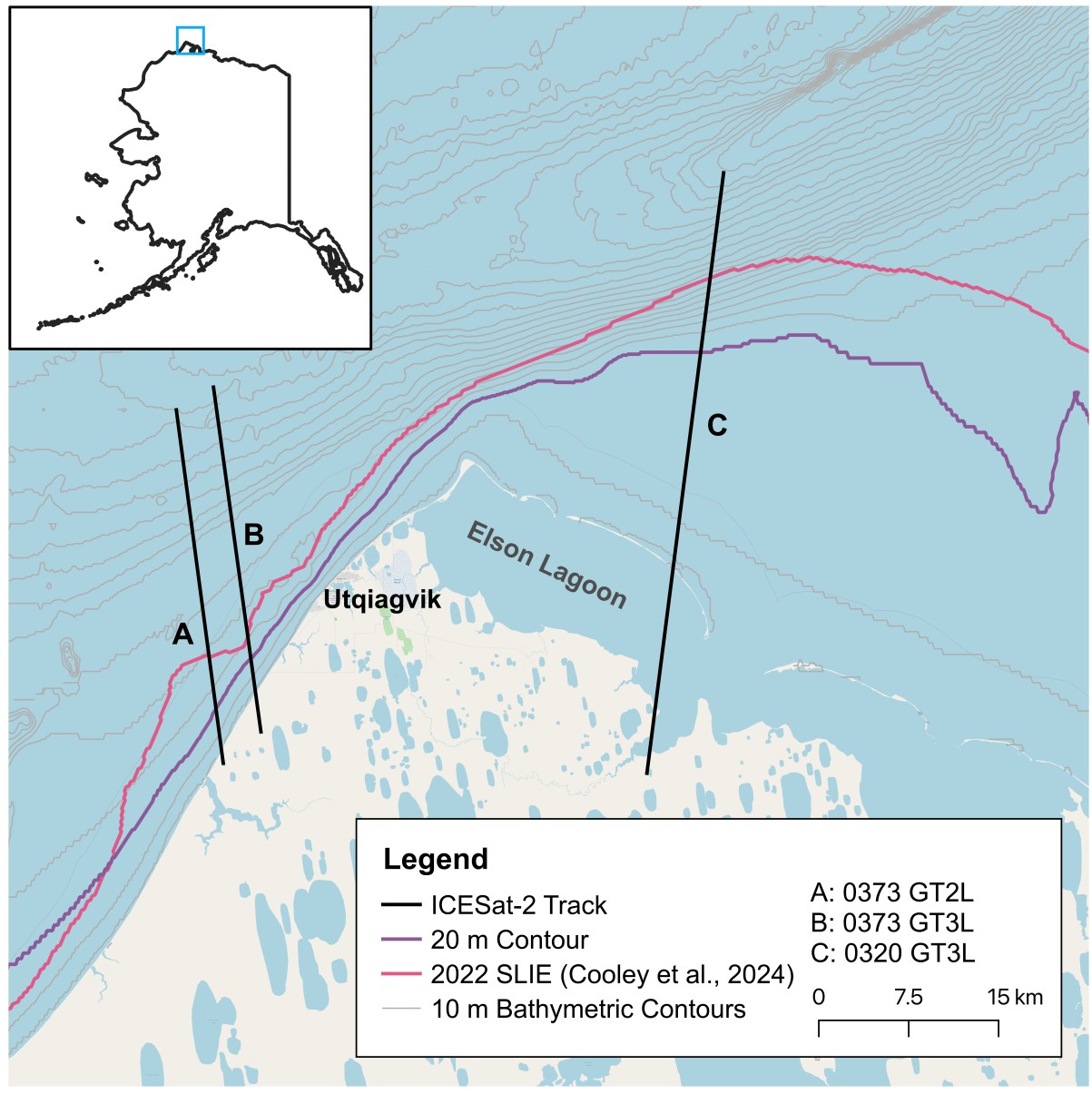

**Figure 2.** An overview of the pilot study area offshore of Utqiaġvik, AK, and the three ICESat-2 ground tracks (GT) labeled A-C (black lines) highlighted in the results. Bathymetric contours (grey lines) are located every 10 m, including the 20 m isobath outlined in purple

along the roughly 1,500 km of coastline, we derive insights into the characteristics of grounded sea ice ridges and how they vary geospatially.

In this study, we analyzed ICESat-2 (Ice, Cloud, and Land Elevation Satellite 2) and Global Elevation Bathymetric Chart of the Oceans (GEBCO) data over the 2021-2022 winter to detect grounded ridges. Sentinel-1 data further validates that the associated features are stationary and can better constrain the timing of ridge formation.

## 2.1 ICESat-2 retrievals

ICESat-2 is a NASA Earth observing satellite, launched in 2018, equipped with an Advanced Topographic Laser Altimeter System (ATLAS), a six-beam laser altimeter arranged in three pairs of strong and weak beams, spaced 3.3 km apart. The satellite measures the topographic height of Earth's surface with a footprint resolution of 11 m spaced 0.7 m apart (Magruder et al., 2024). This instrument is the only satellite-based altimetry system that operates at a high-enough spatial resolution to detect individual pressure ridges (Farrell et al., 2020).

We use the University of Maryland Ridge Detection Algorithm (UMD-RDA, Duncan and Farrell (2022)) to extract surface height from ICESat-2's Level 2A Geolocated Photon Data product (ATL03) between December 2021 and May 2022. By filtering background photons, the algorithm detects sea ice height along each track (Duncan and Farrell, 2022). Over level ice surfaces, such as refrozen leads, ICESat-2 surface heights have a precision of 1-2 cm (Kwok et al., 2019; Farrell et al., 2020). The ICESat-2 UMD-RDA results were validated using independent near-coincident airborne laser scanner data which showed strong correlation (Duncan and Farrell, 2022) but showed that the algorithm underestimates total sail height by $\sim$0.1 m (Ricker et al., 2023; Farrell et al., 2020). For the purposes of this analysis, underestimating sail height slightly is preferable to overestimating; with this magnitude of error we are not likely to miss too many grounded ridges.

We began with 16 and 27 ICESat-2 tracks that crossed 20 km of the Chukchi coastline and 30 km of the Beaufort coastline during the 2021-2022 winter season. Because the satellite orbits on a 91-day repeat, there are a maximum of two passes per winter season over the same location. In the Alaska Arctic, these repeat passes occur in January and April or February and May. However, we found the May passes in the Beaufort pilot tracks return very little data, likely due to persistent fog and/or inclement weather. As a result, there were only seven pairs of repeat satellite passes between the two pilot areas that returned sufficient data to detect individual ridges on both dates, intersected with shore in the study area to provide near-shore measurements, and had two tracks within the period of continuous ice cover to determine whether ridges persisted over the 91 day period. We relied heavily on repeat tracks for initial method development, to assess how the stability of landfast ice was changing over the winter, and for algorithm validation. Because of the limited number of repeat tracks, we use individual tracks for the wider analysis of the coastline.

ICESat-2 tracks often measure ridges at an angle that is not perpendicular to the ridge axis. For consistency, we present ridge locations in terms of the distance from shore rather than distance along track for each individual ICESat-2 surface height measurement. This approach allows us to correct for angular distortions and collect more consistent width data. We are still limited by the unknown angle of the ridge though, and rely on near-shore ridges being on average roughly parallel to the shoreline/bathymetric contours.

## 2.2 Synthetic Aperture Radar

The low temporal resolution of the ICESat-2 product requires a higher temporal resolution product to track the development of ice features in time. Sentinel-1 is a set of satellites, launched in 2014 and 2016 as part of the ESA Copernicus program, equipped with a C-band Synthetic Aperture Radar Instrument that covers the pilot sites once every 12 days. We used Level 1 Sentinel 1A Ground Range Detected (GRD) vertical polarization synthetic aperture radar data, collected in Interferometric Wide mode (ESA, 2022), with a spatial resolution of 20 x 22 m. This was accessed through the Alaska Satellite Facility. Although the spatial resolution is too low to identify most pressure ridges, we use the relative brightnesses to identify persistent features as a proxy for ice stability and therefore infer the presence of grounded ridges (Jones et al., 2016). In doing so, we can narrow down the timing of formation and breakup of grounded ridges to a 12 day window, providing information on sea ice stability and persistence on a seasonal scale. We use Sentinel-1 SAR imagery to validate the stability of ice features surrounding the grounded ridges detected in the ICESat-2 retrievals.

## 2.3 Bathymetry

In order to determine if a ridge is grounded, it is necessary to have a measure of the sea floor depth. For this analysis, we use the Global Elevation Bathymetric Chart of the Oceans (GEBCO) product (GEBCO Compilation Group, 2023), a dataset that combines available bathymetric surveys regardless of sampling methods, and is gridded to a 15 arc-second interval grid ($\sim$ 500 m). While higher resolution data products exist (e.g., NCEI Digital Elevation Model Global Mosaic at a grid resolution of 1/3 arcsec $\sim$ 10 m), we find the higher resolution both limits the speed and extent of processing while also appearing poorly interpolated in some areas of coastal Alaska. The gridded GEBCO product is sampled along the ICESat-2 tracks with a 2D linear interpolation at the horizontal position of each altimetry measurement.

Beyond the relatively coarse spatial resolution of the gridded product, bathymetric data in nearshore areas inherently contains uncertainties due to limited boat access in shallow marine environments. Typical survey approaches may not work, leading to reliance on interpolations from sea floor measurements conducted further offshore in comparison with land based topographic measurements (Amante, 2018). Surveys underlying GEBCO product in shallow water near northern Alaska are primarily measured by single beam echo sounder.

Wind-driven variations in sea level are much greater that than the tidal amplitude (~20 cm near Utqiaġvik (NOAA, 2010)), often exceeding 1 m (Mahoney et al., 2007b). Tides and wind-driven variations in sea level represent a large fraction of the total depth in shallow water. Moreover, near-shore areas are subject to active sediment transport and redistribution, influenced by wave action, coastal erosion, and scouring from grounded ice. Tidal variability and wind-driven sea level changes are fundamentally temporary phenomena. We assume the bathymetric depth represents an accurate time-averaged sea floor depth and recognize the depth uncertainty must be considered in the likelihood that a ridge keel reaches the sea floor in any given location.

## 3 Analysis

We define sea level relative to the retrieved ice surface height based on the thickness of undeformed ice estimated using a freezing degree day model (Anderson, 1961; Maykut and Untersteiner, 1986; Kaleschke et al., 2012). Using linear regressions on a dataset of surveyed ridge geometry from locations in the Chukchi, Bering, and Beaufort Seas (Appendix, Strub-Klein and Sudom (2012)), we estimate the keel depth from each sail height for each point in the UMD-RDA ICESat-2 retrievals. By comparing the estimated keel depths to the depth of the sea floor interpolated from bathymetric data, we can determine whether the ridges are likely to be grounded.

This approach permits a characterization of grounded ridge geometry and location along Arctic coastlines. We process data from 1271 ICESat-2 tracks (527 in the Chukchi Sea, and 744 in the Beaufort Sea) that intersect the coastline between Point Hope, AK and Komakuk Beach, Canada between December 1st, 2021 and May 1st, 2022. For each identified ridge, we quantify grounded ridge height, bathymetric depth, distance from shore, and grounding width.

### 3.1 Freezing degree day model to estimate level ice thickness

An accurate estimate of keel depth relies upon a precise alignment of water level relative to the sea ice surface height measurements (Figure 3). The empirical ratios between sail height and keel depth are based on a freeboard measurement. Without reliable lead detection nearby on a particular ICESat-2 track, we estimate the level ice height from ICESat-2 modal surface retrievals. We then calculate undeformed thermodynamically-grown sea ice freeboard using a freezing degree-day (FDD) model for ice thickness, and determine water level assuming buoyant equilibrium relative to the modal ice height.

We modeled FDD ice thickness using daily average temperature data for five locations across the Alaska Arctic: Kotzebue, Utqiaġvik, Komakuk Beach, Kaktovik/Barter Island, and Prudhoe Bay (NOAA, 2024). These sites provide daily temperature data during the period of October 1, 2021 to May 1, 2022 and are spaced along the entire coastline. Equation 1 calculates the number of freezing degree days ($\Theta$) from the freezing point of seawater ($T_f = -1.9$ C) and the hourly 2 m air temperature ($T_a$) integrated over time (t) from an initial freeze up date ($D_0$) to the date of the measurement ($D_1$).

$$\Theta = \int_{D_0}^{D_1} (T_f - T_a)dt \tag{1}$$

The ice thickness result depends upon the start date of the model, which is the date in which ice forms that persists through the winter season. Finding this date is complicated because early season ice is prone to breakout events. Starting the model too late would underestimate the ice thickness. Detailed community-based ice observations near Utqiaġvik record the beginning of the landfast ice growth season with "stationary ice along shore" on November 15th, 2021 (Observers Billy Adams and Joe Leavitt, Adams and Observers of coastal Arctic Alaska (2022)), but documented landfast ice formation dates are not available across the entire Arctic Alaska coastline. Integrating FDD data from a range of possible start dates ($D_0$ = October 15, November 1, and November 15) provides a likely range of ice thicknesses given the observed weather patterns.

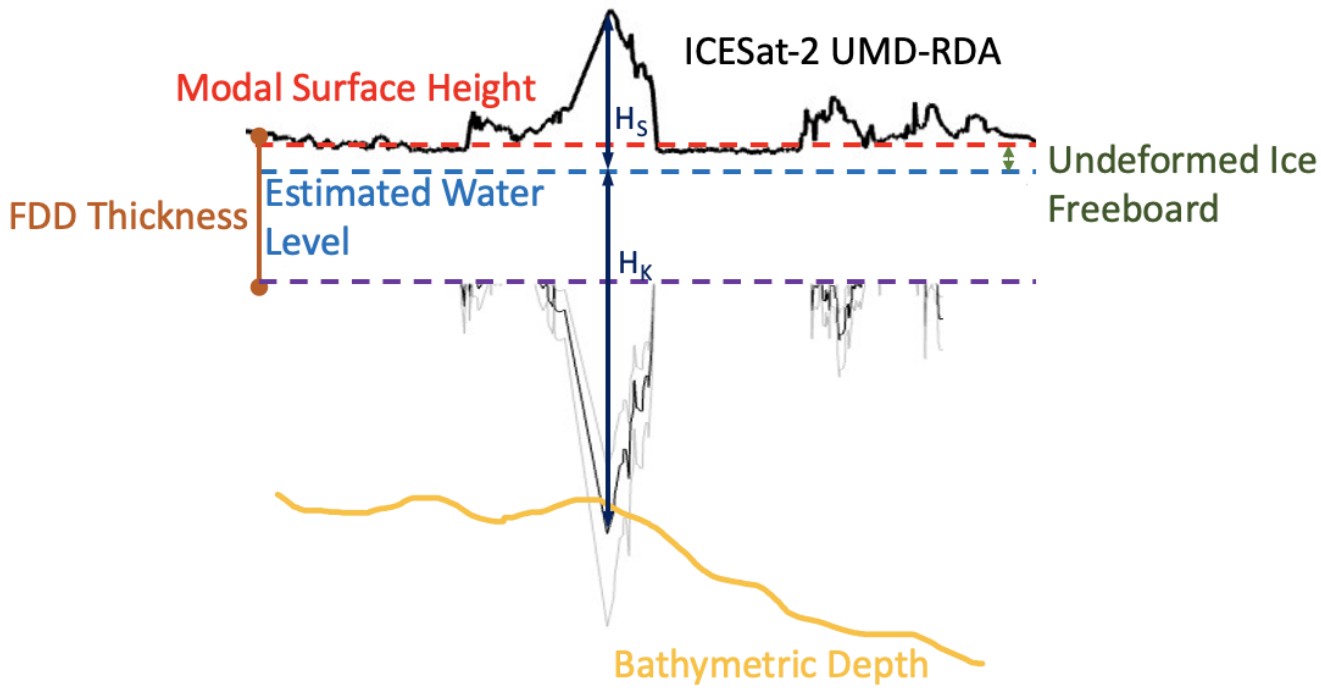

**Figure 3.** Water level for each ICESat-2 track is calculated by assumming buoyant equilibrium and an ice thickness from a FDD model. Subtracting the undeformed ice freeboard from the modal surface height gives a water level that we use to define sail height ($H_s$) and estimate keel depth ($H_k$).

$$H^2 + 5.1H = 6.7\Theta \qquad (2)$$

$$H = 1.33\Theta^{0.58} \qquad (3)$$

Equations 2 (Anderson, 1961; Maykut and Untersteiner, 1986) and 3 (Kaleschke et al., 2012) calculate ice thickness (H) using two different FDD models. Late in the season there is little difference between the models, but the estimated ice thicknesses vary by up to 10 cm in January. For example, with a start date of November 15, we calculated a thermodynamic ice thickness of 84.4 cm and 90.1 cm on January 15 2022 and 146.4 cm and 146.5 cm on April 16 2022 for Equation 2 and Equation 3 respectively.

## 3.2 Sail-keel depth relationship estimates

We rely on empirical equations linking ridge sail height to keel depth from field observations. Strub-Klein and Sudom (2012) published a database of ridge measurements including 147 first-year ice ridges surveyed between 1976 and 2008 in areas near the Alaska Arctic. Ridge locations are indicated by basin, and we consider ridges in the Beaufort and the Chukchi/Bering (grouped in the dataset) regions.

Determining whether a near-shore ridge is grounded depends on the relationship between sail height ($H_s$) (known from ICESat-2 height measurements) and keel depth ($H_k$). Because ridge sizes are influenced by a number of factors including floe momentum and the thickness of the parent ice floe, this ratio varies with geographic location (Strub-Klein and Sudom, 2012). A Wilcoxon test p-value of 0.01 indicates a statistically significant difference in the empirical relationship between sail height and keel depths for ridges in the Chukchi/Bering and Beaufort Seas, accounting for positively skewed trends in the data (Gehan, 1965). Therefore, separate ratios must be calculated for ridges in the Chukchi/Bering and Beaufort Sea regions.

The measured sail height and keel depths are plotted on Figure 4 for each of the Chukchi and Beaufort regions, with the mean ratio and 1 standard deviation on either side to represent the range of possible depths for each ridge. Following Strub-Klein and Sudom (2012), Chukchi ridges are described by $H_k = 3.94H_s$, while ridges in the Beaufort Sea follow $H_k = 4.72H_s$. These results yield that for ridges with the same sail heights, keels in the Beaufort Sea tend to be slightly deeper. We use the standard deviation of 1.99 for Chukchi ridges and 1.78 for Beaufort ridges to provide the upper and lower bands for the keel depth estimate. Figure 4 shows this covers the expected range of keel:sail ratios in the observed ridges, all of which come from non-coastal ice. Without evidence to suggest that keel:sail ratios would be substantially lower in coastal areas, we assume that these ratios indicate which ridges extend deep enough to reach the sea floor.

The presence of snow may impact ridge alignment with water level. Given the presence of overlying snow along each ICESat-2 profile, using a FDD thickness estimate relative to the modal ice surface height may place the water level higher than the true value. Should this occur, ridge sail heights are underestimated from the true value, and ridge depths would therefore be smaller. For example, a sail height that is underestimated by 10 cm would result in an underestimation of keel depth by 39 - 47 cm based on mean keel ratios in the Chukchi and Beaufort Sea, respectively. Given this uncertainty, we acknowledge the possible under detection of grounded features using our algorithm. The ridges most likely to be missed because of snow cover are the low confidence detections with the smallest sails, which would in turn have the smallest grounded depths.

## 3.3 Ridge detection

We apply four criteria to identify grounded ridges: 1) the sails must be at least 0.7 m tall to distinguish from ice floe deformation, 2) sails obey the Rayleigh criteria which distinguishes individual ridges by requiring that the ridge surface descend at least 50% towards the minima on either side of the peak of the peak, 3) water depth must be greater than 3 m to eliminate bottomfast ice, and 4) ridges are greater than 10 m wide and less than 500 m wide. The application of sail height and Rayleigh criteria are consistent with Duncan and Farrell (2022). Additionally, applying a minimum depth eliminates uncertainty in nearshore detections, and applying selection criteria to ridge widths eliminates erroneous detections in the retrievals. To minimize over-

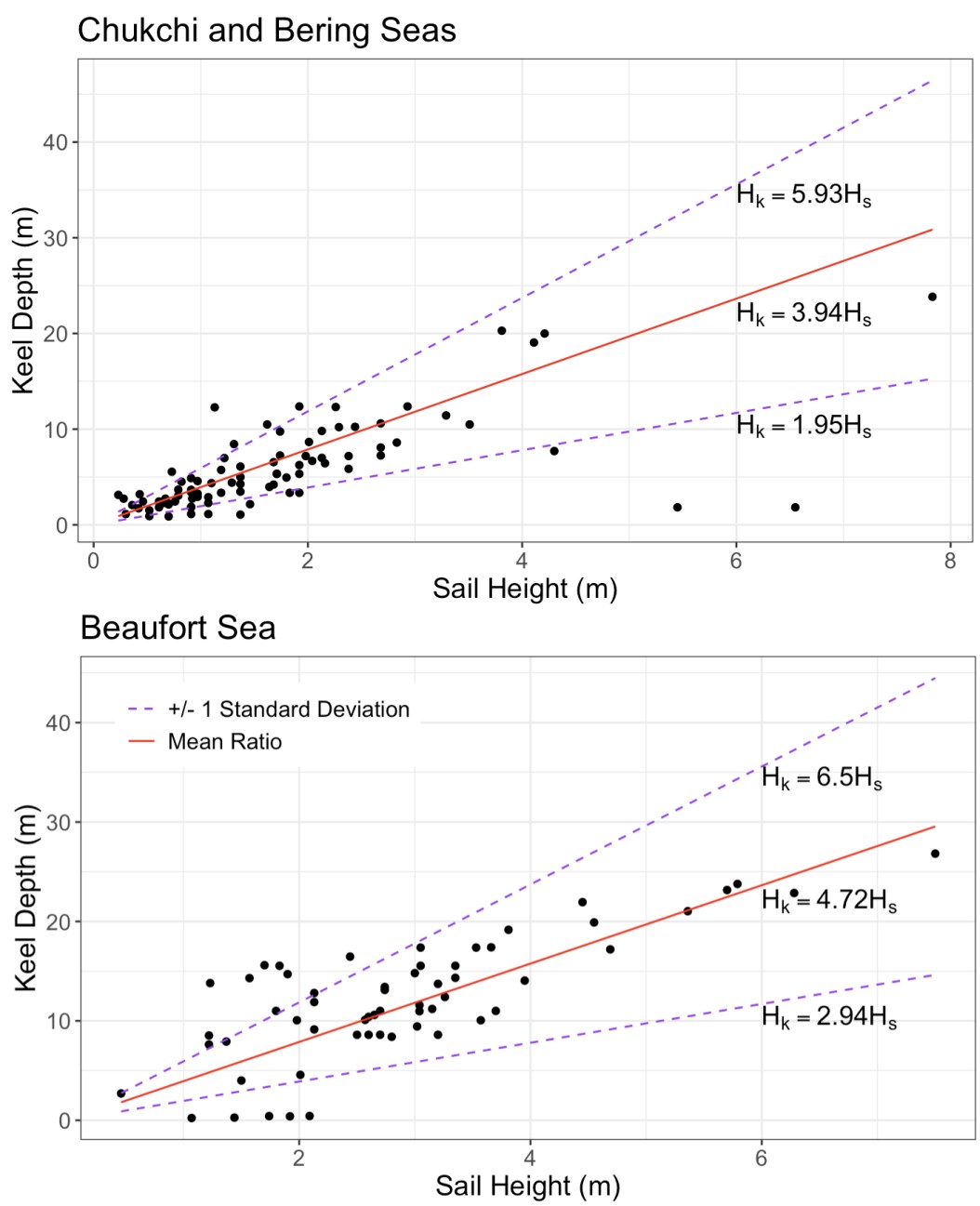

**Figure 4.** Sail heights ($H_s$) and keel depths ($H_k$) from surveyed ridge data (Strub-Klein and Sudom, 2012) in the Chukchi/Bering Seas and the Beaufort Sea with best fit ratio (red) and +/- 1 standard deviation (purple) plotted.

representation in highly deformed areas, we require 200 measurement locations (140 m) between ridge detections. Grounded

ridges are any ridge that meets all of those criteria and where the estimated keel depth is deeper than the bathymetry at that
location.

Detected ridges are assigned to three different confidence levels: high, medium, or low depending on the keel:sail ratio
required to cross the bathymetry level. For instance, the mean ratio minus one standard deviation would result in the shallowest
keel depth estimate. If this conservative depth estimate still intersects the bathymetry, this ridge is classified as a high confidence
ridge. We note that this approach omits calculating an uncertainty value associated with the water level estimate and the
bathymetry calculations: without a significant validation campaign for both the likely ridge geometries and the bathymetric
maps, a more quantitative approach to uncertainty estimation would be inherently incomplete.

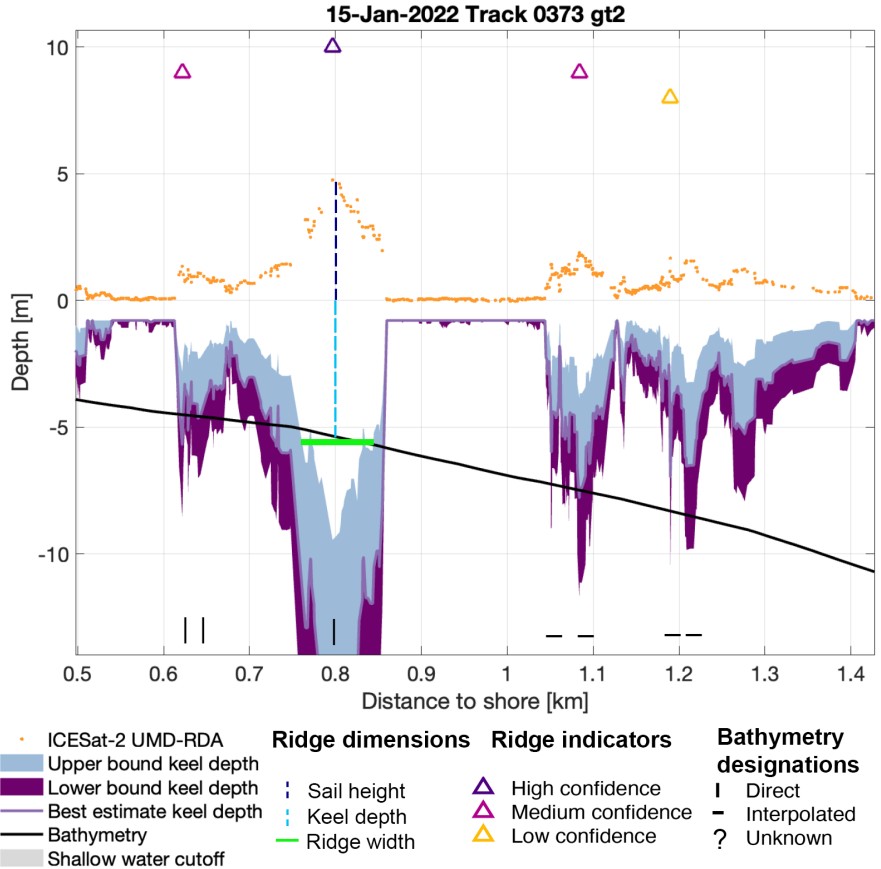

**Figure 5.** Segment of ICESat-2 surface height (orange) across the sea ice in the Chukchi Sea with estimated keel depth (blue/purple) and
bathymetric depth (black). Dark purple indicates the deeper bound of the keel:sail ratio estimates, and blue indicates the shallower bound.
The solid green line indicates the grounded ridge width and the dashed blue lines indicate sail height (dark) and depth (light). The largest of
these ridges is classified as a high-confidence grounded ridge, with other ridges in this section being low- or medium-confidence.

Figure 5 shows one high-confidence ridge, two medium-confidence ridges, and one low-confidence ridge that are detected using our grounded ridge detection algorithm. For the largest of these features, all three of the ratio depth estimates intersect the bathymetry, while for the other ridges, the most conservative depth estimate (small ratio) does not. We record several characteristics of the identified grounded ridges that will provide insights into cross-coastline dynamics, labeled on Figure 5. The height is based on the peak of each ridge detection, and depth is measured at this same distance from shore. We measure the width of the grounded section of the ridge: the full width is where the keel depth estimate crosses below the bathymetry line, measured as distance to shore. For ridges that are not oriented parallel to shore, this approach overestimates the width of the ridge.

It is important to note that the estimated keel profiles are not meant to be an accurate portrayal of the under side of the landfast ice. An empirical relationship between maximum sail height and maximum keel depth only applies to ridges and does not estimate the underside of undeformed sea ice. Ridge keels are typically wider than ridge sails by a mean factor of 6.75 (with a standard deviation of 4.8 up to an observed maximum of 35, Strub-Klein and Sudom (2012)), and this is not accounted for in the keel depth estimates or in the ridge grounded width measurement. We have applied the FDD modeled ice thickness estimate to undeformed ice so that the figures in this paper show only estimated deformation that extends deeper than the thermodynamic growth. We also neglect ridge keel width as focusing on the maximum depth is sufficient for determining whether each ridge is grounded.

Finally, we calculate the location of each ridge relative to the March/April 2022 landfast ice edge as detected by Cooley and Ryan (2024), recording it as a percentage between the SLIE and coastline (Equation 4). This provides some insight into where the landfast ice is being stabilized.

$$\frac{\text{percent out}}{\text{from coast}} = \left( \frac{\text{distance to coast}}{\text{distance to coast} + \text{distance to SLIE}} \right) \times 100\% \tag{4}$$

## 4 Results

We apply this approach first to three pairs of repeat tracks in the pilot study area to show the detection of persistent grounded ridges, the formation of new grounded ridges during a season, and how the characteristic ridge geometries differ between the Chukchi and Beaufort sides of the Alaska coast (location indicated on Figure 2). We then apply the ridge detection methodology to the entire northern Alaska coastline for a broader study of ridge characteristics.

We consider the two sets of repeat tracks on the Chukchi side of Utqiaġvik and one set on the Beaufort side to demonstrate the capabilities of the algorithm in (1) detecting persistent grounded ridges, (2) detecting changes in grounded ridges, and (3) comparing the Chukchi and Beaufort Sea ice deformation characteristics. We use repeat tracks to confirm that ridges identified as grounded in the early-winter track (January) are still in place in the late-winter track (April), and more generally, to better understand when grounded ridges are forming during the winter season. When we identify immobile landfast ice in both the January and April ICESat-2 tracks as indicated by persistent ridges and in the SAR imagery, we have evidence that the ice is stabilized off-shore by a grounded ridge.

## 4.1 Detection of persistent grounded ridges

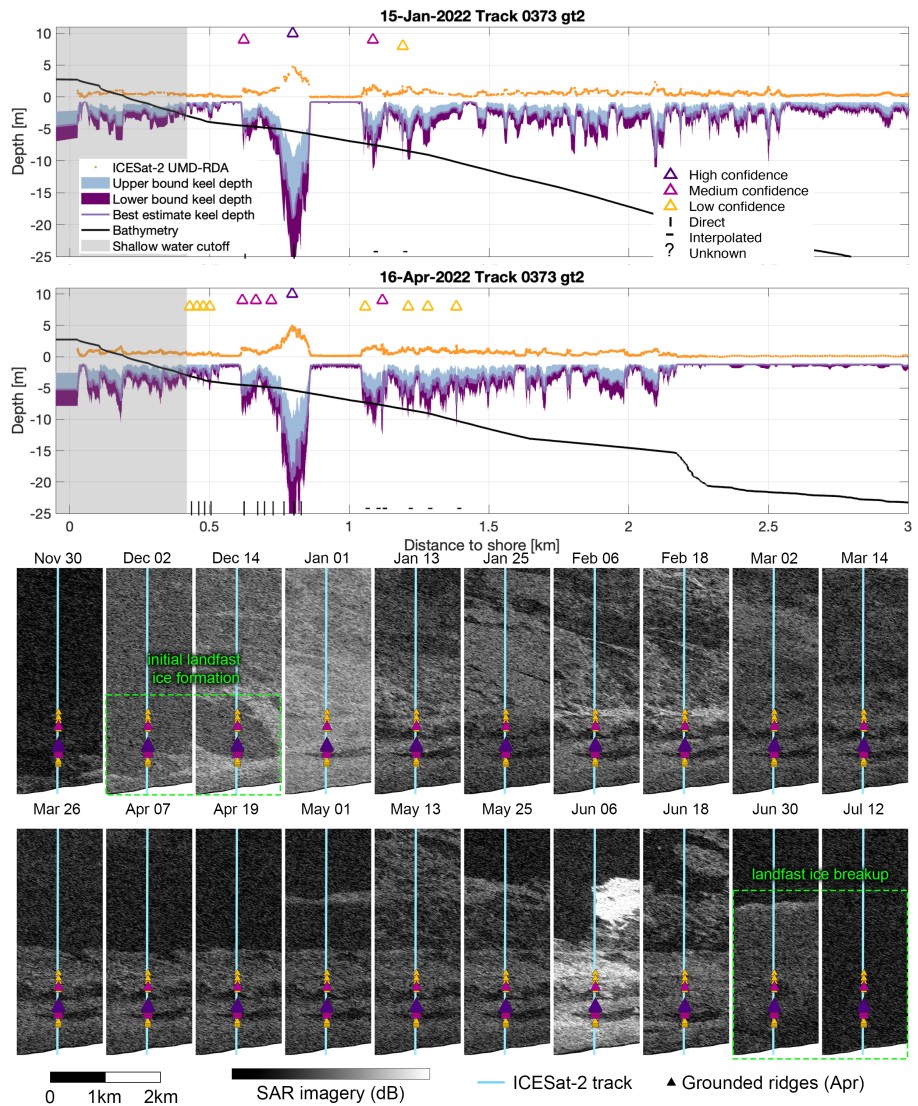

**Figure 6.** Top: Paired ridge profiles from ICESat-2 tracks 0373 GT2L in the Chukchi pilot area to the west of Point Barrow (71.19° N, 157.04° W) on 15 January and 16 April 2022. Ridges detected as grounded are noted using yellow (low confidence), pink (medium confidence), or purple (high confidence) markers. Symbols on the bottom row indicate direct (|), interpolated (-), or unknown source (?) bathymetry at the locations of grounded ridges. Bottom: Sentinel-1 SAR backscatter images surrounding the ICESat-2 track (0373 GT2L, indicated by the cyan line) with the location of grounded ridges identified in the April ICESat-2 track indicated by triangles matching the above color scheme. Panels outlined in green denote the periods of formation and breakup of persistent features in the sea ice.

Figure 6 shows the surface height of the ice from ICESat-2 using the UMD-RDA (orange) along ground track 0373 in the Chukchi Sea and the GEBCO interpolated bathymetric depth (black). Estimated keel depths are shown in the purple and blue bands; the keel depth estimate based on the best linear fit of the surveyed ridges is labeled in light purple and the dark purple shading showing low confidence estimates on the relationship between sail height and keel depth and blue shading showing high confidence depth estimates. The blue shading represents the most conservative depth estimate based on our sail height:keel

depth ratios, so we consider ridges high confidence when this top of this estimate band intersects the bathymetry. Each ridge is marked according to it's highest-confidence detection. The row at the bottom of the figure indicates the source of the bathymetry estimate: we use the vertical line to indicate grounded ridge locations based on direct bathymetry measurements and horizontal lines for areas of interpolated bathymetry. This provides insight into the uncertainty of bathymetry surrounding each grounded feature, with directly measured bathymetry likely to better represent the sea floor location. The slight differences in bathymetric

profile between the two tracks are a result of close but not quite overlapping ground tracks of the ICESat-2 repeat surveys.

   Using these depth estimations, our algorithm returned one high confidence ridge that intersects with the sea floor between 0.75 km and 0.85 km from shore. This feature persists from the January 15 track through the April 16 track. Based on the keel estimate and the persistence of the feature, this seems very likely to be a grounded ridge. Additionally, one medium confidence ridge persists between the two dates around 1.1 km from shore. Imagery from the Utqiagvik Sea Ice Radar (UAF, 2022) shows

persistent features in the area at similar distances from shore during this time.

   In the SAR imagery, we note persistent features in the ice around the ICESat-2 track that form between December 2 and December 14 last until the end of June. Because these features, including the dark band that is slightly shoreward of the grounded ridge (purple dot), remain unchanged throughout the season, we can infer the stability of the ice and confirm the presence of grounded ridges in the area. The spatial resolution of the SAR imagery poses a challenge for unambiguous interpretation of

grounded ridges and is an insufficient tool by itself for determining where exactly ridge features form or for characterizing their geometry. Leveraging the additional context from the SAR imagery, we can confirm the grounded ridge detections from ICESat-2 altimetry can monitor identified ridges over the season from formation to breakup.

   We note that surface features shore-ward of this grounded ridge in Figure 6 remain largely unchanged between the tracks, indicating that the landfast ice is stable over this 91 day period. Surface topography on the seaward side differs more between

the two tracks. This is also apparent in the SAR imagery where a new seaward landfast ice edge (SLIE) is established between March 14 and March 26th. There are some differences in the detected ridges between January and April: gaps in the altimetry data in January mean that the feature at 0.6 km km is not classified as a separate ridge in January, but with no gaps in the April retrievals, it is considered a separate ridge. With the keel depth estimates not accounting for keel width, whether this was actually a separate ridge is an open question.

Since the altimetric surface height profile suggests that the grounded ridge sail at 0.8 km from shore is ∼100 m wide, we expect that it is a combination of rubble from the collision(s) that formed the ridge, drifted snow, and likely runs slightly off from parallel to shore. This is consistent with widths assumed in Mahoney et al. (2007a). The ICESat-2 track passes roughly perpendicular to shore in this area, and it appears from a time series of Sentinel-1 SAR imagery (Figure 6, lower panels) that the ridged feature near 0.8 km from shore is approximately perpendicular to the altimetry track. The feature in the SAR imagery

appears to be on the order of 50-200 m wide in the off-shore direction and more than a kilometer long in the along-shore direction.

## 4.2 Feature persistence and detection of new grounded ridges

The period from January to April between repeat tracks is more than long enough for the development of new grounded ridges. Figure 7 shows a series of 4-5 low/medium confidence ridge features between 0.4 km and 1.45 km off shore that are likely grounded from before January 15 through the April 16 ICESat-2 tracks. The April 16 track includes an additional medium-confidence grounded ridge feature from 1.85 - 2 km off shore. Beyond their estimated depth extending deeper than the local bathymetry, the persistence of the features shoreward of 1.5 km between the two track dates suggests stability.

In the SAR imagery, there are persistent features near where we identify the ridge 1.8 - 2 km from shore that form in late February (after the initial ICESat-2 track data) and last through the May 13 image. The ICESat-2 profile of the ridge shows that the horizontal scale of the feature is $\sim 200$ m wide and the sail extends $\sim 5$ m above the surface with a keel reaching approximately 20 m deep. This depth is consistent with the stamuki zone ($\geq 10$ m) where ridges ground near the landfast ice edge.

In this area, we see evidence of ridges forming at a range of angles relative to the shoreline, both in the SAR imagery (ESA, 2022) and in the Utqiagvik Sea Ice Radar (UAF, 2022). Pack ice movement in the area can be seen moving at a wide range of angles relative to shore (UAF (2022) 2021-2022 winter animations), which would be consistent with these geometries. While this track has only a little overlap with the field of view of the Utqiagvik Sea Ice Radar, there are prominent ridge features evident in the radar at $\geq 0.6$ km from shore and evidence of landfast ice extending out to $\sim 2$ km from shore later in the season (UAF (2022) 2021-2022 winter animations).

Between the January and April tracks, there are slight changes in the best estimate keel depths. This is not likely due to a change in the surface height of the features themselves, but rather that the repeat tracks are not perfect overlaps of the same ground footprints. Snow accumulation and redistribution may also change the detected surface height of the flat, undeformed ice surface relative to the ridges. The ridge peaks in the April tracks are also slightly higher relative to the estimated water level than in the January tracks because the water level is calculated from the buoyant equilibrium of thermodynamically grown ice thickness estimated with the FDD model. As such, we see an increase in the number of low-confidence ridges detected in the 0.6 - 1.5 km from shore area between January and April. These ridges appear in the January track, but were not estimated to be deep enough to be grounded.

## 4.3 Comparison of Chukchi and Beaufort ridge detections

Figure 8 applies the same approach as used in Sections 4.1 and 4.2 to shorefast ice on the Beaufort side of Point Barrow, using the empirical relationship between sail height and keel depth derived for the Beaufort region. Here, we find that the likely grounded ridges are located farther from shore than in the tracks examined on the Chukchi side of Point Barrow, consistent with a wider band of landfast ice (Cooley and Ryan, 2024; Mahoney et al., 2014). This is a result of the shallower sea floor extending farther from shore along the coastline of the Beaufort Sea, as well as a different ice motion regime relative to the

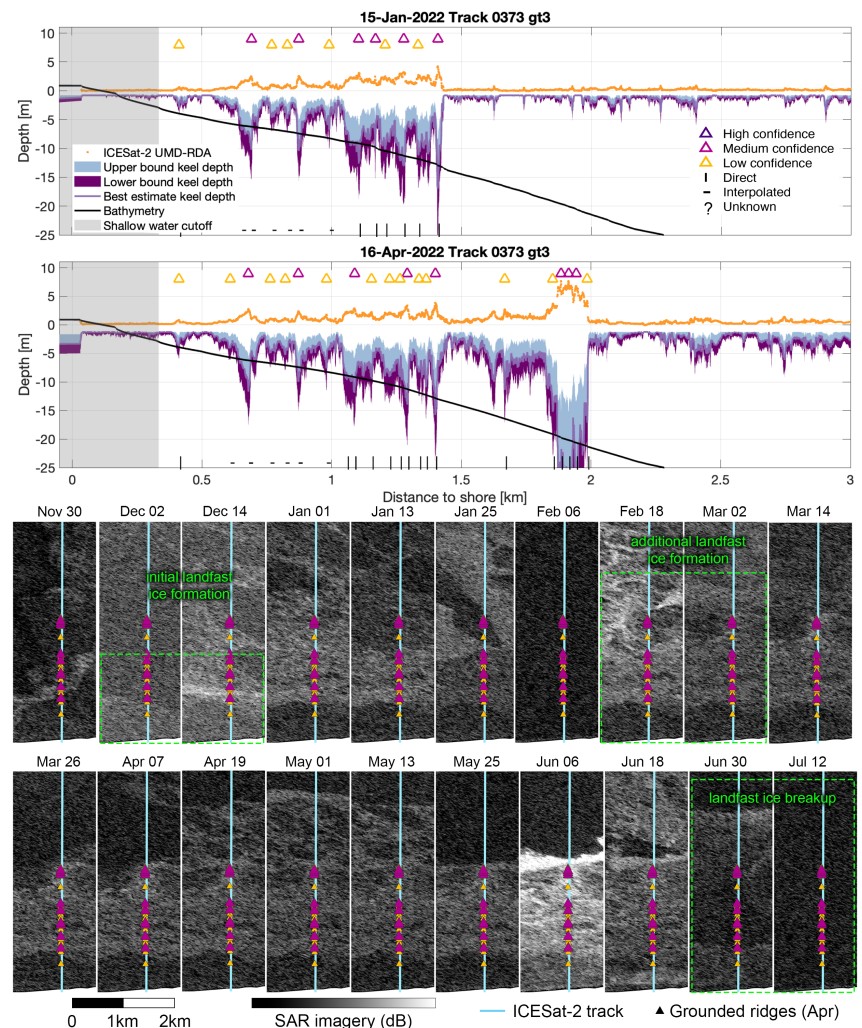

**Figure 7.** Top: Paired ridge profiles from ICESat-2 tracks 0373 GT3L in the Chukchi pilot study area to the west of Point Barrow (71.23° N, 156.96° W) on 15 January and 16 April 2022. Ridges detected as grounded are delineated using yellow (low confidence), pink (medium confidence), and purple (high confidence) markers. Symbols on the bottom row indicate direct (|), interpolated (-), or unknown source (?) bathymetry at the locations of grounded ridges. We find a series of medium confidence grounded ridges ~0.7, 0.85, 1.1, 1.3, and 1.4 km off shore, based on keel depth estimates (purple and blue shading) intersecting the bathymetry (black). An additional large grounded ridge is formed sometime after the January 15 ICESat-2 track around ~1.9 km from shore. Bottom: Sentinel-1 SAR backscatter images surrounding the ICESat-2 track (0373 GT3L, indicated by the cyan line) with the grounded ridge detections indicated by triangles corresponding to the above color scheme. Panels outlined in green denote the periods of formation and breakup of persistent features in the sea ice.

shoreline (Zhao and Liu, 2007). The patterns in ridges persist between the January and April tracks indicate the stable shorefast ice likely extends to at least 6 km, and up to 13 km, from shore.

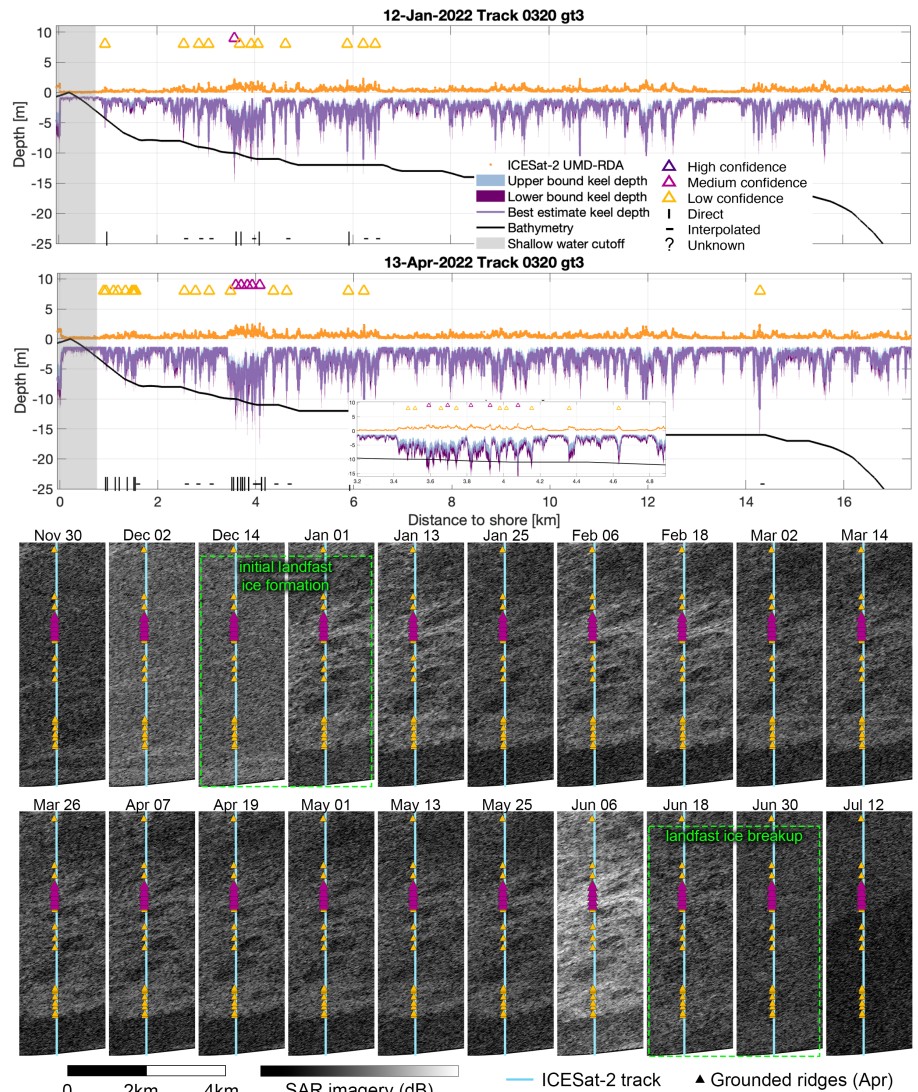

**Figure 8.** Top: Paired ridge profiles from ICESat-2 tracks 0320 GT3L in the Beaufort Sea to the east of Point Barrow (71.32° N, 155.99° W) on 12 January and 13 April 2022. Ridges detected as grounded are delineated using yellow (low confidence), pink (medium confidence), and purple (high confidence) markers. Symbols on the bottom row indicate direct (|), interpolated (-), or unknown source (?) bathymetry at the locations of grounded ridges. We find an area with possibly grounded ridges between 3.5 - 4 km off shore, shown in the inset, based on keel depth estimates (purple and blue shading) intersecting the bathymetry (black). Bottom: Sentinel-1 SAR backscatter images surrounding the ICESat-2 track (0320 GT3L, indicated by the cyan line) with detected ridges shown as triangles. Panels outlined in green denote the periods of formation and breakup of persistent features in the sea ice.

Our examples illustrate that the ice surface in the Beaufort Sea study area is notably more deformed than the Chukchi Sea study area, with smaller ridges (<2 m sail height and <50 m width) occurring roughly every 200 m. In this profile, there are not larger ridges (∼5 m sail height), which we found in the two Chukchi-side tracks.

We note several persistent features in this track, but do not identify any as high confidence grounded ridges. The nearshore section may have ∼0.5 km of bottomfast ice in the shallow (<3 m) water near the spit separating Elson Lagoon from the Beaufort Sea. We note one feature with significant deformation around 3.5–4.2 km from shore that persists between the January and April tracks but only intersects the bathymetry line for the medium and low-confidence ratios. Further features out to 13 km from shore are highly similar between the two tracks, and the SAR imagery suggests little ice movement between January 1 and June 6. We suggest, based on the consistency of the ice features over the 91 day period, that this region of shorefast ice may be laterally stabilized by grounded ridges.

## 4.4 Ridge characterization and mapping

Applying this algorithm across the entire Alaska Arctic, we measure 773 individual grounded ridges in the Chukchi Sea along ICESat-2 tracks, and 1669 along the Beaufort coastline (2442 total grounded ridges) that are classified as high, medium, or low confidence. We compile a database of parameters for each ridge including height, depth, width, and distance from shore for each ridge.

The statistics (Table 1) reveal several notable differences between ridges in the Chukchi and Beaufort regions. We find that grounded ridges in the Beaufort Sea tend to be wider, deeper, and taller than those in the Chukchi Sea, while the number of grounded features per track is comparable between the two locations. Across the entire coastline, high confidence retrievals tend to have a greater mean ridge height and width, while are located at shallower depths and closer to shore than lower confidence retrievals.

Grounded ridges in the Beaufort are located on average 8.07 km for high confidence, 7.13 km for medium, 6.17 km for low confidence detections farther from the shoreline than the grounded ridges in the Chukchi Sea. This is expected; bathymetric conditions control the grounding location of landfast ice and depth increase more quickly along the Chukchi coastline than they do along the Beaufort coastline.

Our data shows mean depths of $6.1 \pm 2.9$ m in the Chukchi and $7.3 \pm 3.9$ m in the Beaufort regions for medium confidence ridges, with only 11.6% of all Chukchi ridges and 27.9 % of all Beaufort ridges falling within the stamuki zone ($\geq 10$ m bathymetric depth). Only 1.7% and 8.9% high confidence ridges were detected at depths greater than 10 m, and 8.8% and 18.7% for medium confidence ridges, and 13.9% and 33.5% for low confidence ridges in the Chukchi and Beaufort Seas, respectively. Our results indicate that ridges ground the sea ice at all bathymetric depths between the landfast ice edge and the shoreline, with a majority occurring in shallower water. Only 6 Chukchi ridges and 64 Beaufort ridges were located at depths greater than 20 m, the approximate landfast ice edge location. Even if we only consider the furthest seaward ridge along each track, the average depth is still only 6.7 m and 8.3 m in the Chukchi and Beaufort regions, respectively, for medium confidence ridges. This disconnect between the traditional definition of the grounding zone along coastlines and our results indicates one of a few things: climatic changes and a warming Arctic are causing grounded ridges to form at shallower depths, more shoreward grounded ridges have been ignored, and/or the bathymetric data is incorrect.

Barker and Timco (2017) found that grounded ridge sails along Arctic coastlines form up to 15 m in height. This is consistent with all ridge heights detected using this method.

| Region | Confidence | Metric | Height (m) | Width (m) | Depth (m) | Distance (km) |
|---|---|---|---|---|---|---|
| Chukchi | High | min | 1.8 | 11.0 | 3.0 | 0.03 |
| | | mean | 3.4 | 56.9 | 4.4 | 1.06 |
| | | max | 7.6 | 293.7 | 11.8 | 8.40 |
| | | stdev | 1.3 | 56.6 | 1.8 | 1.44 |
| | Med | min | 0.8 | 10.6 | 3.0 | 0.39 |
| | | mean | 2.3 | 40.4 | 6.1 | 1.91 |
| | | max | 7.6 | 401.4 | 20.2 | 13.60 |
| | | stdev | 1.2 | 63.8 | 2.9 | 2.24 |
| | Low | min | 0.7 | 10.7 | 3.0 | 0.40 |
| | | mean | 1.7 | 26.5 | 7.3 | 3.65 |
| | | max | 5.2 | 190.6 | 3.0 | 22.52 |
| | | stdev | 1.1 | 66.8 | 3.1 | 3.25 |
| Beaufort | High | min | 1.1 | 10.9 | 3.0 | 0.02 |
| | | mean | 3.0 | 34.7 | 6.0 | 9.13 |
| | | max | 7.5 | 214.5 | 17.0 | 24.05 |
| | | stdev | 1.4 | 31.6 | 2.9 | 7.33 |
| | Med | min | 0.7 | 10.8 | 3.0 | 0.09 |
| | | mean | 2.1 | 25.9 | 7.3 | 9.04 |
| | | max | 7.0 | 282.9 | 22.8 | 54.93 |
| | | stdev | 1.3 | 42.1 | 3.9 | 7.54 |
| | Low | min | 0.7 | 10.2 | 3.1 | 0.23 |
| | | mean | 1.9 | 23.6 | 9.3 | 9.82 |
| | | max | 6.6 | 127.7 | 35.0 | 70.74 |
| | | stdev | 1.2 | 38.9 | 4.9 | 8.30 |

**Table 1.** A summary of calculated grounded ridge metrics from the Chukchi and the Beaufort Seas from December 1, 2021 - May 1, 2022. The full list of grounded ridges with their dimensions is provided in the supplemental material.

Mahoney et al. (2007a) assumes that ridges deep enough to become grounded were on the order of 100 m wide. Our average sail widths are 40.4 and 25.9 m for medium confidence Chukchi and Beaufort grounded ridges, respectively. Even when considering the furthest seaward ridge along each track, mean ridge widths over both coastlines are still 40.6 m for high confidence, 39.4 m for medium confidence, and 30.8 m for low confidence ridges. We suggest this difference may result from the method by which ridge widths are detected in our algorithm – individual ridge peaks in larger rubble fields are recorded instead of the full width of highly deformed ice and our measured width only accounts for the part of the sail tall enough to result in a grounded keel depth.

Our results suggest, on average, 1.9 and 2.5 medium-confidence grounded ridges per ICESat-2 ground track in the Chukchi and Beaufort regions respectively (with ranges of 0.4 - 5.1 and 0.5 - 6 .7 between high and low confidence ridges in the two basins). Given these tracks pass over an effectively random sampling of coastal Arctic sea ice, and that there are on average >1 ridge per track, we can infer that any along-shore section of ice is likely grounded by at least one ridge in the vicinity, though there are certainly some places without any grounded ridges (approximately 16% of ICESat-2 surface height profiles examined contained no grounded ridges at any confidence level, and 40% had no high-confidence grounded ridges). This suggests that coastal sea ice is regularly supported by frequent grounded ridges in the shallow water (< 10 m depth) and infrequent grounded ridges in the deeper stamuki zone. This is a higher spatial density in the shallow water zone than in pack ice (Mahoney et al., 2007a). If the primary mechanism for grounded ridge formation was the advection of deep-keeled ridges from the pack ice, we would expect a similar spatial density. This would suggest a combination of in situ ridge formation and advected ridges contributing to the grounding of landfast ice in the shallow water zone.

We use the landfast ice edge, detected by Cooley and Ryan (2024) using ice stability measured from MODIS data from March and April, 2022, to consider where in the landfast ice we find grounded ridges. We find that the average high confidence grounded ridge location was 23% and 17% of the distance between the coastline and landfast ice edge in the Chukchi and Beaufort Sea regions, respectively, located much closer to the shoreline than to the SLIE.

Figure 9 shows probability plots comparing the grounded ridge measurements, enabling a comparison of grounded ridge geometry across the two coastlines. From this, insights are derived into geospatial variability of ridge geometry that may result from varying bathymetry or directions of predominant ice drift.

We find ridges are more likely to be 2-3.5 m tall in the Chukchi region, while ridges are relatively more likely to be greater than 4 m in height in the Beaufort region. Beaufort ridges are more likely to occur at depths greater than 10 m than those located along the Chukchi coastline. Due to regional bathymetry, ridges are more likely to be located within 2 km of shore in the Chukchi Sea, while ridges in the Beaufort are more likely to be located at greater distances from shore. Finally, Chukchi ridges are more likely to have widths greater than 50 m, while Beaufort ridges tend to have smaller grounded widths. Though we do not measure individual non-grounded ridges in this study, ICESat-2 profiles show much more deformed ice on the Beaufort side but the ridges are less likely to be particularly large or grounded. Higher prevalence of deformed ice along the Beaufort coast is consistent with Strub-Klein and Sudom (2012).

In the Chukchi Sea, grounded ridges at the same bathymetric depth tend to be slightly taller than those in the Beaufort region. This difference in ridge height may be attributed to a combination of factors, including shoreline orientation relative to

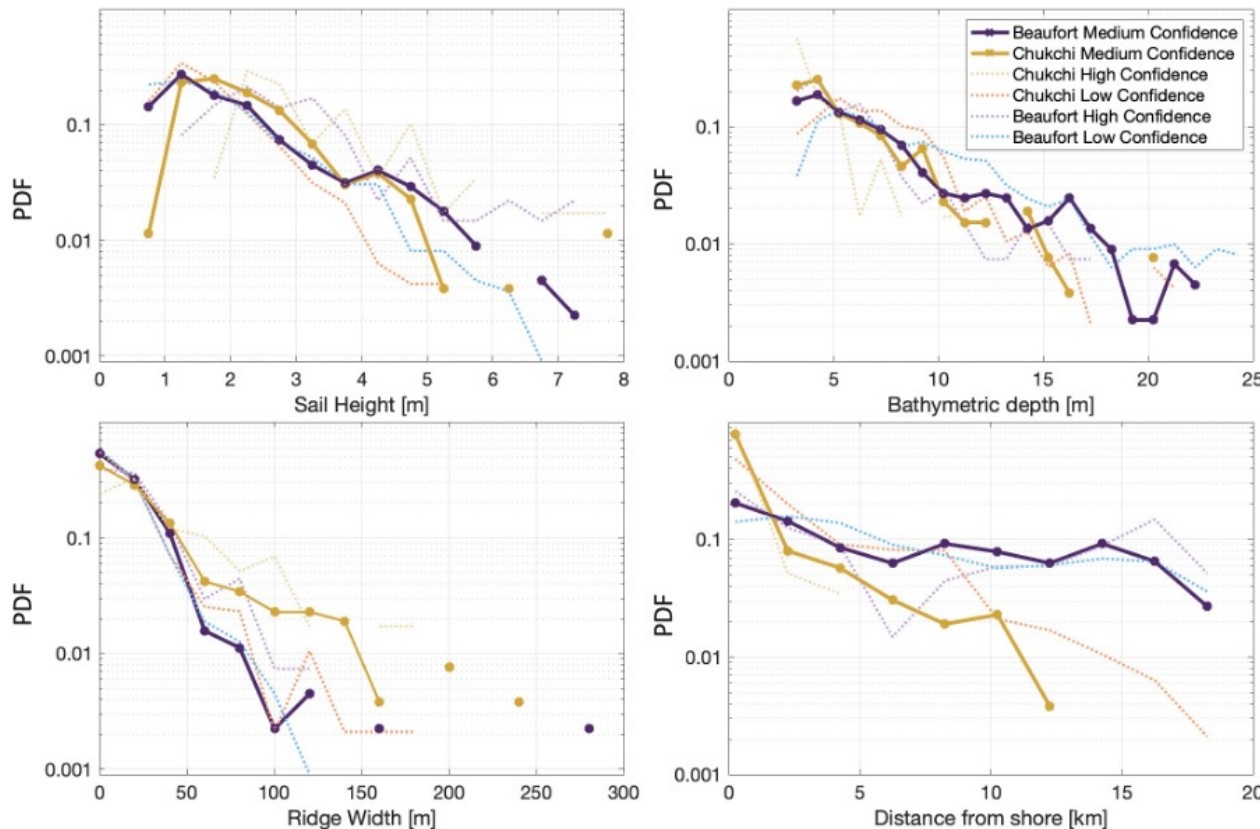

**Figure 9.** Probability density function (PDF) for ridge height, depth, width, and distance from shore. The bold yellow (Chukchi) and purple (Beaufort) lines show medium confidence ridges, with the dotted yellow and purple lines showing high confidence ridges and orange and blue indicating low confidence detections.

predominant ice drift, the thickness of the parent ice floe, and the process by which the ridges form (advection versus in situ deformation). When the sea floor limits the keel depth, a larger amount of total deformation could result in either a taller sail, wider ridge, or both. Parsing these mechanisms would require a more detailed process study of keel formation.

Next, we show that seasonal trends in grounded ridge sail height, depth, distance from shore, and width can be delineated from the data with infrequent repeat measurements over the same geographic area. We plot means of grounded ridge properties over two week periods starting December 1st on Figure 10 and find that height, depth, width, and distance from shore tend to increase until April. We plot the average of all detected ridges on the left panels, and perform the same calculation using only the ridge located furthest from shore along each ICESat-2 track on the right panels.

For all grounded ridges in the Chukchi Sea region, the mean sail height more than doubles during the winter season, from 0.9 m in December 2021 to 2.3 m by the end of April 2022. A similar seasonal trend is present in Beaufort sail heights, increasing from 1.5 m in December to 2.2 m in April. The bathymetric depths of grounded ridges in the Chukchi and Beaufort regions

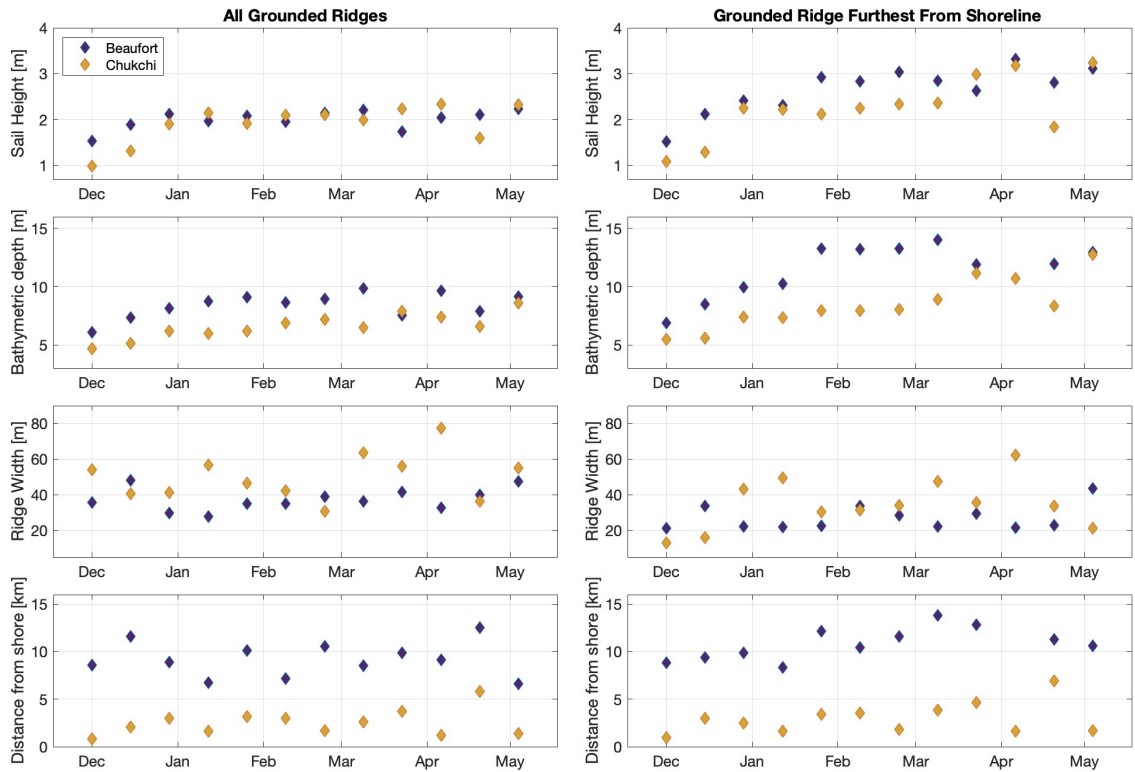

**Figure 10.** Seasonal evolution of grounded ridge sail height, bathymetric depth, grounding width, and distance from shore between 1 Dec 2021 and 1 May 2022 for all ridges, regardless of confidence level. The right column includes only the ridge located furthest from shore along each ICESat-2 track. Data were aggregated over 2-week periods, with each period including a number of ridges ranging from 9 to 111 in the Chukchi Sea and 42 to 261 in the Beaufort Sea.

increased similarly over the season, from 4.7 m in December to 7.9 m in mid-March, and from 6.1 m in December to 9.9 m in end of February respectively, forming at shallower depths in the Chukchi Sea throughout the season. The mean distance from shore and width calculated based on all grounded ridges is variable throughout the season, but there are more evident trends when considering only the grounded feature furthest from shore along each track.

We find a much larger increase in sail height when accounting only for the grounded ridge furthest seaward along each ICESat-2 track, increasing to mean heights of 3.2 m and 3.3 m for Chukchi and Beaufort ridges, respectively in early April. A larger increase in depths is observed over the season for the seaward ridges, increasing from 5.5 m to 11.2 m in the Chukchi Sea and from 6.9 m to 14.0 m in the Beaufort Sea. Mean ridge width increases from 12.9 m in early December to 62.2 m in early April along the Chukchi coast, and from 21.3 m in December to 43.5 m by May along the Beaufort coast. Finally, ridge distance from shore increases from 0.9 km to 6.9 km along the Chukchi coast and from 8.8 km to 13.8 km along the Beaufort coast for the furthest seaward ridge.

Through a comparison of ridge dimensions for all ridges and those of the feature located furthest from the coastline on each ICESat-2 track, we can gain insight into landfast ice evolution and its stabilizing mechanisms. We propose that these increases in height, depth, width, and distance from shore reflect ongoing collisions and ridge-forming events that both build upon existing features and create new ridges at increasing distances from shore, and hence greater depths. These trends suggest that subsequent collisions not only increase the height of existing ridges but also form additional ridges at greater depths throughout the season. Seasonal trends in distance from shore are complicated by the fact that each track contains ridges formed at any time up to that point during the season—for example, a March track might include near-shore ridges formed in January. We interpret these seasonal increases as evidence that ridges form further from shore and at greater depths as shore ice stabilizes, and that subsequent collisions build up ridges, causing them to increase in size over the season. The latter would suggest an increase in width over the season: given increases in width for seaward ridges are only slightly larger than those for all ridges, it seems likely that more of the winter changes in distance from shore are due to the formation of new ridges further from shore (e.g., Section 4.2).

Figure 11 maps grounded ridge locations. Extended linear features form roughly parallel to the shoreline, as we would expect from frequent pack ice drift towards the southwest in the southern Beaufort Sea. Notably, one distinct feature spans approximately 150 km between -146 and -150°W (near Kaktovik, Alaska), roughly parallel to the shoreline. While ICESat-2 data alone cannot determine whether these are extensions of the same grounded ridge or separate shorter ridge segments, this feature highlights the potential for this approach for a broader study of the conditions and locations in which grounded ridges tend to form.

Nodes from Mahoney et al. (2014) are plotted on Figure 11, representing locations of recurring grounded ice features where the landfast ice edge persistently occurs across years. Grounded ridges reported in this study are present in all nodes. Three specific nodes are reported in greater detail on Figure 11A, Figure 11B, and Figure 11C. On node A, we find ridges located further from shore than the surrounding coastline, and located upon the landfast ice edge from this winter. Node B is located approximately at the landfast ice edge, though does contain a number of ridges that likely anchor the ice further from the shoreline. Finally, Node C follows a notable feature that extends parallel to the shoreline. This is the highest density of grounded ridge detections - with a wide swath of detections, it is unclear whether these are extensions of the same ridge, or whether this is an area of frequent deformation and many subsequent collisions.

Along the Chukchi coast, ridges are located close to the shore, with a mean distance for all reported ridges of 2.9 km off shore. In contrast, along Beaufort Sea coastline, grounded ridges are generally located much farther from shore, with a mean distance of 9.6 km. Nearly all of the grounded ridges detected in this study fall within the bounds of the independently-estimated landfast ice edge for March/April 2022 (blue, which is only shown west of 143° W due to the available data from Cooley and Ryan (2024)).

We also plot the height and depth of ridges across this coastline (Figure 11, lower panels), finding that patterns in regional variability across the coastline are difficult to distinguish. The overall statistics indicate that ridges form at slightly deeper locations in the Beaufort region compared to the Chukchi, but there is significant variability in ridge height and distance from shore in both areas. Relatively few ridges fall at depths greater than the 20 m isobath on either side of the coast.

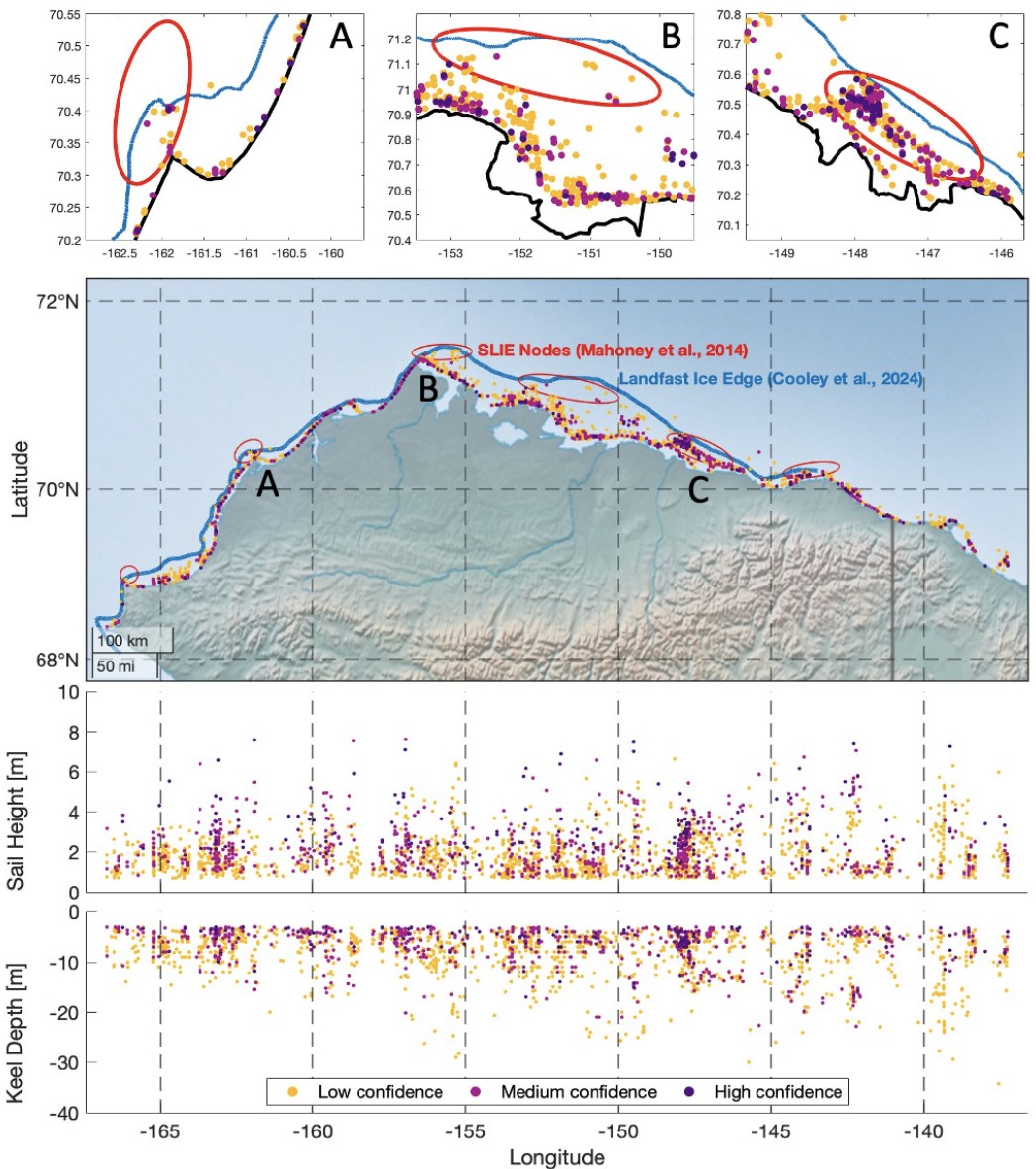

**Figure 11.** Grounded ridges detected in ICESat-2 tracks (purple, pink, yellow for high - low confidence detections) along the Alaska Coastline between December 1, 2021 and May 1, 2022. The blue line represents the landfast ice extent as published by Cooley and Ryan (2024) and the red circles represent interannual locations of the landfast ice edge published by Mahoney et al. (2014)

There are a few notable gaps in the data, including around longitudes 159° W, 142° W, and 139° W. These are gaps in ICESat-2 coverage, likely due to adverse weather on one or both of the two passes in these locations over the winter season.

## 4.5 Limitations of this approach

There remains some uncertainty around the role of relatively flat, undeformed ice in some of the assumptions underlying our analysis. Sail heights are measured relative to the surface of the surrounding undeformed ice, but the calculation of keel depth is based on the height of the sail relative to the water line. Thermodynamic ice growth over the course of the winter season will slightly change the water line relative to ridge peaks and accordingly change the estimate of the sail height ($H_s$) that we use to calculate the keel depth. If the change in thermodynamic ice thickness is very large, we would expect repeat tracks to show artificial growth of the ridge keels over the winter season. There are slight differences in keel depth between repeat tracks (e.g., Figure 8), but the differences are small enough that they may be a result of the slight differences between the track footprint on one overpass versus another.

Detections of grounded ridges are likely impacted by tidal and wind-driven sea level changes. Assuming the level ice can float independently, positive variations in sea level would lift the level ice around the grounded ridge, thereby reducing the relative height of the ridge sail ($H_s$), and leading to an underestimate of keel depth ($H_k$). If the level ice is more firmly connected to the grounded ridges, the ridges themselves could be lifted off the sea floor given a large enough storm surge. We acknowledge that it is possible that grounded ridges within the scope of this study are undetected due to sea level variability, but do not expect there is a systematic bias in under-detecting ridge heights in a particular region. With thousands of kilometers of ICESat-2 retrievals during the 2021-2022 winter, we do not expect that the exclusion of some ridges will significantly change our final results.

There is also some uncertainty associated with the bathymetric data used in our study: sediment transport, ice scouring, and limitations of measuring depths close to the coast in icy waters all mean that the bathymetric data represents an approximation of the depth of the sea floor. For ridges where all depth estimates intersect the sea floor, and the features remain persistent between two dates, we can detect grounded ridges with some certainty.

Our analysis of ICESat-2 repeat-track data throughout winter 2021–2022 suggests that the repeat alignment of the ground tracks in this area is accurate to 10-20 m. This is within the mission requirements (Markus et al., 2017) and consistent with the ICESat-2 footprint size of ∼11 m (Magruder et al., 2024). However, given the small-scale variability of sea ice ridges and rubble, small changes in ground-track geolocation between repeat passes could result in differences in pressure ridge height and width. Analyzing the difference in surface height between pairs repeat tracks did not yield notable results and could be explained by snow redistribution or subsequent collisions causing ridges to shift slightly.

Finally, it is possible some of the features which we record as grounded are not actually ridges in the sea ice. Oil rigs are a known infrastructural feature prevalent offshore from the Alaska coastline, as part of the National Petroleum Reserve. They are likely to be present in some of the ICESat-2 tracks, and due to the resolution of the altimetry product, and the size of the rigs, we suspect they could appear similar to a ridge without further context. Frequent spring fog in the region (Khalilian, 2016) may also cause erroneous surface elevation measurements resulting from reflected photons which are not captured by the filters in the data processing algorithm.

## 5 Conclusions

Using available data from ICESat-2, Sentinel-1, and GEBCO bathymetric products, we present a methodology to identify grounded sea ice ridges over the course of a winter season. This approach can be applied to coastal regions throughout the Arctic, wherever there is a need for monitoring the development of grounded ridges. This process can be combined with additional data products (e.g., high-resolution radar or SAR imagery) to confirm the stability of surrounding ice features. Comparing surface heights from ICESat-2 repeat tracks in the Chukchi and Beaufort Seas, we could confirm that particular ridges were grounded and stabilizing the surrounding ice. For communities with either infrastructure to support a sea ice radar system (e.g., Utqiaġvik, (UAF, 2022)) or with active community observers (e.g., Alaska Arctic Observatory and Knowledge Hub database (Adams and Observers of coastal Arctic Alaska, 2022)), these surface-based observations could be used instead of or in addition to the SAR imagery for constraining ridge development in time. While underlying uncertainties in bathymetric data exist, these can be addressed with additional bathymetric surveys in near-shore environments.

These findings are significant for a number of reasons: grounded ridges impact coastal community subsistence hunting, transportation, and safety, provide habitat for animals, and stabilize the coastline from erosion. In the face of warming temperatures and the uncertainties of climate change, our approach may serve as a critical tool for ongoing monitoring and understanding of shorefast ice stability and seasonality. As we accumulate a longer ICESat-2 dataset, this approach may be used to track changes in shorefast ice timing and grounded ridge formation over time.

To continue improving on our approach and addressing some of the known limitations, in situ surveys of ridge morphology in the near-shore environment are needed to further validate this process. We rely on statistics of ridge geometry in off-shore environments to estimate keel depths, which may not necessarily apply to ridges in nearshore ice. Additional bathymetric surveys are necessary to constrain uncertainties associated with sea floor depth estimates.

A wider application of this approach to shorefast ice observation could be used to validate modeling efforts at capturing landfast ice dynamics (e.g., (Lemieux et al., 2015)). Grounded ridge formation occurs at sub-grid scale in current sea ice models, though these ridges impact larger scale ice movement and near-shore dynamics. Modern models are incapable of reflecting interannual landfast ice variability which impacts sea ice thickness and concentration results, halocline stability (Itkin et al., 2015), upwelling estimations (Kasper and Weingartner, 2015), and brine expulsion (Selyuzhenok et al., 2015). Shorefast ice also blocks momentum flux between the atmosphere and ocean, isolating the ocean from wind-driven mixing and limiting upwelling, and causes river plumes to extend to further distances under the ice (Granskog et al., 2005). In particular, the suggestion that most grounded ridges are far shoreward of the SLIE means that the cohesion/tensile strength of sea ice may be an important part of coastal dynamics (as in König Beatty and Holland (2010)). Ultimately, the approach outlined in this study creates opportunities to observe grounded ridge properties across much larger spatial scales, opening potential avenues for improved understanding of coastal Arctic processes.

*Code and data availability.* The processing code used for this project is available at https://doi.org/10.5281/zenodo.15014586 . ICESat-2 ATL03 data are publicly available from the National Snow and Ice Data Center (https://nsidc.org/data/atl03/versions/6). The UMD-RDA processed data used in this study is available at https://doi.org/10.5281/zenodo.12188016 .

*Author contributions.* KL: data analysis, writing, AB: project conception, advising, writing, SLF: ICESat-2 data processing, reviewing of draft/writing, KD: ICESat-2 data processing

*Competing interests.* No competing interests are present.

*Acknowledgements.* SLF and KD were supported by NASA Cryosphere Grant 80NSSC20K0966. KL was supported by student research funding through the Williams College Geoscience Department.

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
