# Peer review of "Grounded ridge detection and characterization along the Alaska Arctic coastline using ICESat-2 surface height retrievals"

_EGUsphere, 2024_

## Referee Comment (RC2)

**Grounded ridge detection and characterization along the Alaskan
Arctic coastline using ICESat-2 surface height retrievals**

by Kennedy A. Lange, Alice C. Bradley, Kyle Duncan, and Sinéad L. Farrell

*Submitted to The Cryosphere Discussions*
*https://doi.org/10.5194/egusphere-2024-1885*

**Review**
*Aug 30, 2024*

**Summary**
   This manuscript presents an analysis of the distribution of grounded sea ridges near the Alaska coast in the Chukchi and Beaufort Seas based on a combination of ridge sail heights derived from ICESat-2 altimetry data and GEBCO bathymetry data. The approach can be boiled down to 3 key steps. First, the individual ridges are identified and their sail heights determined using ICESat-2/ATLAS geolocated photon height (ATL03) data following the University of Maryland Ridge Detection Algorithm (Duncan and Farrell, 2022). Second, the range of likely keel depths associated with each ridge sail is estimated using sail height/deel depth ratios derived data reported by Strub-Klein and Sudom (2012). Lastly, grounded ridges are identified as those with keel depths exceeding the water depth at their location. The result of this approach is an impressively high fidelity picture of grounding locations within the landfast ice during the 2021-22 winter that suggests grounded ridges for in deeper water in the Beaufort Sea than the Chukchi, but overall the vast majority of grounded ridges are found in waters less than 15 m deep.

   Overall, I enjoyed reading this paper and I believe the work will make a valuable contribution to our understanding of landfast stability and sea ice / seafloor interaction. However, in preparing this review I identified several concerns relating to a lack of methodological detail, missing discussion of relevant physical processes, under-utilization of uncertainty calculations, and perhaps an incomplete reading of some of the cited literature. Being quite familiar with landfast ice in this region, I find the sparsity of grounded ridges near the 20-m isobath surprising, particularly in the Beaufort Sea, and in its current form, I feel the manuscript leaves me with too many questions to take such surprising results at face value.

   I want to stress that I would very much like to see this work published. I don't think any of my concerns should be too difficult to address, and once they are, I would be feel more comfortable about accepting the authors finding regarding the water depth distribution of grounded ridges. I have provided detailed comments below explaining my concerns and I have attempted to make constructive recommendations for improving the manuscript.

**Major Comments**

1. Why are the keel/sail ratios different from those reported by Strub-Klein and Sudom?
   The principle findings of this manuscript critically depend on the ratio between sail height and keel depth. I am therefore surprised to see no discussion regarding the difference between the ratios determined via the linear regressions illustrated in Figure 3 and those reported by Strub-Klein and Sudom. For the Chukchi Sea, the authors find a linear regression with a slope of 3.37 and confidence intervals with slopes of 2.92 and 3.82, while Strub-Klein and Sudom report a mean ratio of 3.92. In the Beaufort Sea, the difference is greater with Fig 3 showing a regression slope of 3.49 and confidence interval slopes of 2.74 and 4.25 while Strub-Klein and Sudom report a mean value of 4.72. Given the

significance of these ratios in the determining the apparent presence or absence of grounded features, I feel these differences should be discussed in some depth. For example, how many more grounded features would be identified beyond the 10 m isobath if the authors used Strub-Klein and Sudom's ratios?

2. Uncertainties in keel depth and water depth are discussed but under-utilized
    The authors list a number of sources of uncertainty that could affect their findings and on lines 131-132, the text states "*These uncertainties are acknowledged in this study and the results are interpreted accordingly*". I commend the authors for including confidence intervals on many of their figures, but the only subsequent reference to them is on lines 382-383 where the text reads "*For ridges where the whole 95% confidence interval (indicated by the purple shading) intersects the sea floor, and the features remain persistent between two dates, we can detect grounded ridges with some certainty*". Aside from this brief and rather qualitative statement, I can find no other indication that the confidence intervals are taken into account when identifying grounded ridges. As a result, the principal way in which the authors account for uncertainties in keel depth and bathymetry appears to be simply acknowledging their existence and the occasional use of qualifying terms like "potential" and "possible" when referring to the identification of grounded ridges. Instead, I encourage the authors to consider assigning some quantitative level uncertainty to each grounded feature based on the degree of overlap between the confidence intervals for keel depth and bathymetry. At the least, the authors need to clarify whether or not the confidence intervals play any role in the identification of the grounded ridges illustrated in Figure 10.

3. Some additional clarification regarding derivation of sail height would be useful
    The text states that sail heights are measured relative to the surrounding undeformed ice, the freeboard of which is estimated based on a freezing degree day model. However, there is no specific explanation regarding how the surrounding undeformed ice is defined or identified. Duncan and Farrell refer to height of the local level ice surface ($H_L$), which "is computed as the mode of the $h_c$ height distribution in 25 km along-track segments", where $h_c$ is the height of the sea ice surface above the mean sea surface, corrected for tides and atmospheric conditions. These are important details for interpreting the results presented in this manuscript and I feel they should be included in the text so that the reader does not have to search a separate publication. I would also like to see an explanation of how these 25-km track segments are treated at the coast. Specifically, are they are truncated at the coastline and, if so, what effect might this have on the derivation of level ice height and, therefore, ridge sail heights? Please also see comment 4 about the impact of sea level variations on sail height measurement for grounded ridges.

4. Some further discussion of local variations in sea level may be required
    Although line 129 makes references to a 20-cm tidal range near Utqiagvik and acknowledges this can be a significant fraction of the water depth, I feel further discussion is required. First, I feel the text should recognize that wind-driven variations in sea level are much greater that than the tidal amplitude and can exceed 1 m (e.g. Mahoney et al, 2007b as cited in the manuscript). More significantly however, variations in sea level don't just affect water depth. They can have a much greater effect of the estimation of keel depth due to the way in which sail heights of grounded ridges are measured. Unlike floating ice, the height of grounded ridges relative to the surrounding ice will vary with local sea level and I think this is a more likely explanation than changes in snow depth for the "*slight changes in the best estimate keel depths*" noted on line 258. For example, the sails of grounded ridges will appear lower when the sea level (and the surrounding non-grounded ice) rises. This would have then have an amplified effect on the estimated keel depth: a 20-cm rise in sea level would reduce the sail heights of

grounded ridges by 20 cm, which would then reduce the estimated keel depths by more than 60 cm, depending on the keel/sail ratio used. Hence, the number of grounded ridges could be significantly underestimated if the ridge sails were measured during a surge in local sea level and the authors might consider correcting their sail heights to account for the dynamic topography of the ocean. In effect, the use of the corrected sea ice height, $h_c$, to derive the height of level ice, $h_L$, (as described by Duncan and Farrell, 2022) may be counterproductive for the purpose of estimating sail height of grounded ridges.

5. GEBCO data are of questionable reliability in shallow water
   Given the sparsity of sounding points in shallow water and the year-to-year variability of the bathymetry in these regions, I am somewhat skeptical of the validity of using GEBCO bathymetric data all the way up to the coastline. The GEBCO Type Identifier (TID) Grid, available with GEBCO bathmetry data, show that broad areas of the Alaska coast shallower than ~5 m are coded with the value 41 which indicates the bathymetry values are "Interpolated based on a computer algorithm". I would therefore recommend the authors exercise more caution when identifying grounded features in shallow water. The authors already exclude features in water shallower than the level ice drafter, but I don't understand the rationale for this (see comment 6). Instead, I recommend the authors establish a shallow-water cut-off value, based on the GEBCO TID Grid. This value might vary regionally according to the density of sounding points used in the GEBCO grid, but this would provide a data-driven rationale for excluding ridges in shallow water.
   Also, under the topic of bathymetric uncertainty, I don't understand how the orange areas in Figures 4-7 were derived. Line 209 states "*The orange shading represents positional uncertainty in the bathymetry line*", but the width of the orange shading would suggest that the positional uncertainty is on the order of hundreds of meters, which seems too high.

6. Exclusion of ridges in water shallower than draft of undeformed ice seems unnecessary
   I don't understand the rationale for excluding ridges in water shallower than the draft of undeformed ice. For such ridges to exist at the location, they must have become grounded at some stage when the ice was thinner. Hence, excluding ridges based in undeformed ice draft creates the possibility of a scenario in which a ridge is counted as grounded on one day, but excluded the next. I would therefore discourage the authors from excluding features on this basis and instead simply apply a shallow water cut-off as explained in comment 5.

7. Some more careful reading of Mahoney et al (2007a, as cited in the text) may be required
   The text makes many references to the work of Mahoney et al (2007a), which I feel mischaracterize what is written in the cited work. For example, on line 308, the text states "*Mahoney et al. (2007a) assumes a sail width of 100 m for grounded ridges*", but this is a rather inaccurate description of what was written in that paper. For the purposes of estimating the possible spatial density of grounded ridges, Mahoney et al assumed that the keels of ridges deep enough to become grounded were on the order of 100m wide. This is not the same as assuming grounded ridge sails are 100 m wide.
   Similarly, line 315 begins with the statement "*Mahoney et al. (2007a) suggested grounded ridges discontinuously pin the landfast ice edge, roughly every 30 km*". In this case, I feel the authors are taking what is written in the cited article somewhat out of context. Instead, Mahoney et al. wrote that grounded ridges would be spaced approximately every 30 km if the thickness distribution of landfast ice was the same as that measured offshore in the drifting pack ice. And they go on to note that grounded ridges are observed more closely spaced than this, suggesting that they are produced through "in-situ" grounding process (i.e. like those illustrated in Figure 1 of this manuscript).
   Lastly, on line 320, the text claims "*Mahoney et al. (2007a) assumes that these grounded ridge features are located close to the SLIE*", but I again feel this does not accurately represent what Mahoney

et al. wrote. Rather than making any general assumption that grounded ridges are located close to the SLIE, they identify "nodes" where the SLIE occurs most commonly and suggest that these correspond to the location of interannually recurring grounded features. Additionally, I recommend the authors consider citing Mahoney et al's more recent 2014 paper, which expands on the discussion of these nodes and provides photographic documentation of a grounded ridge in the location of a node that happens to be very close to the location of line C in Figure 2. I also feel that some further discussion of the location of the nodes identified by Mahoney et al in relation to the distribution of grounded features shown in Figure 10 would be a valuable addition to the text.

**Minor comments**

Line 9 (and throughout the text): The word "Alaskan" is a noun referring to someone from Alaska. When used an adjective, the correct term is Alaska.

Figure 1: This figure illustrates one of two ways in which a sea ice ridge can become grounded. The other way involves advection of a deep-keeled ridge into shallow water. Mahoney et al (2007b as cited in the manuscript) speculate that the latter is more likely to create a gouge in the seafloor and is therefore more significant for stabilizing the landfast ice. The authors may wish to acknowledge both ways of creating a grounded ridge and the differences between them.

Figure 2: Why did the authors select these ICESat-2 tracks? Had they selected tracks that intersected the coast closer to Utqiagvik, they could fallen within the footprint of UAF's sea ice radar and there's a good chance that they would have intersected whaling trails where sea ice thickness has been routinely measured since 2007 (https://arctic-aok.org/data-sources/whaling-trail-mapping/). These data are both reference could have provided useful validation for the thickness of level ice in the region. Also, why is Line C truncated before the 20m isobath? From several years of observations, the landfast ice in this region is commonly anchored by grounded ridges just beyond the end of Line C.

Lines 265-267: There are multiple assertions here that should be supported by references. I think I know what the authors mean by the "classic" ridge, but I have a suspicion that its classic status derives from simplifications adopted in many illustrations of coastal ice over the years together with a bias in the early literature toward ridges in the Beaufort Sea. Also, can the authors provide a reference supporting the "prevalance of shear" on the Chukchi Side of Point Barrow? The climatological prevailing wind is from the east, with creates a lot more divergence than shear and a lot less shear than on the Beaufort side of Pot Barrow

Line 337-338: The reasoning behind this explanation in the final sentence of this paragraph is not clear to me. If grounded ridges in the same water depth are higher in the Beaufort Sea than in the Chukchi, this means the ice is piled higher above the waterline in the Beaufort. I can envision a few mechanisms that might cause such a difference (for example in-situ grounding vs advection of deep-keeled ridges in shallow water) and how they might relate to coastal aspect or parent ice thickness, but I feel most readers would benefit from further explanation

---

## Author Comment (AC1)

Grounded ridge detection and characterization along the Alaska Arctic coastline using ICESat-2 surface height retrievals
by Kennedy A. Lange, Alice C. Bradley, Kyle Duncan, and Sinéad L. Farrell
Submitted to The Cryosphere Discussions
https://doi.org/10.5194/egusphere-2024-1885
Review Aug 30, 2024

Responses in blue text.

**Summary**
This manuscript presents an analysis of the distribution of grounded sea ridges near the Alaska coast in the Chukchi and Beaufort Seas based on a combination of ridge sail heights derived from ICESat-2 altimetry data and GEBCO bathymetry data. The approach can be boiled down to 3 key steps. First, the individual ridges are identified and their sail heights determined using ICESat-2/ATLAS geolocated photon height (ATL03) data following the University of Maryland Ridge Detection Algorithm (Duncan and Farrell, 2022). Second, the range of likely keel depths associated with each ridge sail is estimated using sail height/deel depth ratios derived data reported by Strub-Klein and Sudom (2012). Lastly, grounded ridges are identified as those with keel depths exceeding the water depth at their location. The result of this approach is an impressively high fidelity picture of grounding locations within the landfast ice during the 2021-22 winter that suggests grounded ridges for in deeper water in the Beaufort Sea than the Chukchi, but overall the vast majority of grounded ridges are found in waters less than 15 m deep.

Overall, I enjoyed reading this paper and I believe the work will make a valuable contribution to our understanding of landfast stability and sea ice / seafloor interaction. However, in preparing this review I identified several concerns relating to a lack of methodological detail, missing discussion of relevant physical processes, under-utilization of uncertainty calculations, and perhaps an incomplete reading of some of the cited literature. Being quite familiar with landfast ice in this region, I find the sparsity of grounded ridges near the 20-m isobath surprising, particularly in the Beaufort Sea, and in its current form, I feel the manuscript leaves me with too many questions to take such surprising results at face Value.

I want to stress that I would very much like to see this work published. I don't think any of my concerns should be too difficult to address, and once they are, I would be feel more comfortable about accepting the authors finding regarding the water depth distribution of grounded ridges. I have provided detailed comments below explaining my concerns and I have attempted to make constructive recommendations for improving the manuscript.

We thank the reviewer for their kind words and helpful comments which we address individually below in blue text:

**Major Comments**

1. Why are the keel/sail ratios different from those reported by Strub-Klein and Sudom?
The principle findings of this manuscript critically depend on the ratio between sail height and keel depth. I am therefore surprised to see no discussion regarding the difference between the ratios determined via the linear regressions illustrated in Figure 3 and those reported by Strub-Klein and Sudom. For the Chukchi Sea, the authors find a linear regression with a slope of 3.37 and confidence intervals with slopes of 2.92 and 3.82, while Strub-Klein and Sudom report a mean ratio of 3.92. In the Beaufort Sea, the difference is greater with Fig 3 showing a regression slope of 3.49 and confidence interval slopes of 2.74 and 4.25 while Strub-Klein and Sudom report a mean value of 4.72. Given the significance of these ratios in the determining the apparent presence or absence of grounded features, I feel these differences should be discussed in some depth. For example, how many more grounded features would be identified beyond the 10 m isobath if the authors used Strub-Klein and Sudom's Ratios?

We had used a first-order polynomial fit to the sail heights/keel depths in order to account for the thickness of undeformed ice: using a consistent ratio can introduce errors in smaller ridges. That said, those small ridges are (a) likely to be filtered out given the requirement to exceed 0.7m height above modal surface level that comes from the UMD-RDA, and (b) unlikely to be grounded ridges anyways. We have re-run the analysis, using the mean ratios identified in Strub-Klein and Sudom ratios and accounted for the variability in keel/sail ratios in the observations by using a full standard deviation for the range of values. Our new ratios reflect the 3.92 and 4.72 sail:keel ratios for Chukchi and Beaufort seas, with standard deviations of 1.99 and 1.78 respectively. We treat the three ratios for each basin (mean - SD, mean, mean + SD) as 'high', 'medium', and 'low' confidence keel depths respectively.

All of the figures throughout the paper have been updated based on these updated keel depth ratios.

2. Uncertainties in keel depth and water depth are discussed but under-utilized
The authors list a number of sources of uncertainty that could affect their findings and on lines 131-132, the text states "These uncertainties are acknowledged in this study and the results are interpreted accordingly". I commend the authors for including confidence intervals on many of their figures, but the only subsequent reference to them is on lines 382-383 where the text reads "For ridges where the whole 95% confidence interval (indicated by the purple shading) intersects the sea floor, and the features remain persistent between two dates, we can detect grounded ridges with some certainty". Aside from this brief and rather qualitative statement, I can find no other indication that the confidence intervals are taken into account when identifying grounded ridges. As a result, the principal way in which the authors account for uncertainties in keel depth and bathymetry appears to be simply acknowledging their existence and the occasional use of qualifying terms like "potential" and "possible" when referring to the identification of grounded ridges. Instead, I encourage the authors to consider assigning some quantitative level uncertainty to each grounded feature based on the degree of overlap between the confidence intervals for keel depth and bathymetry. At the least, the authors need to clarify whether or not the confidence intervals play any role in the identification of the grounded ridges illustrated in Figure 10.

Thank you for this feedback: we have re-run the analysis detecting grounded ridges from keel depth estimates as described above. We use the three ratios for each basin (mean - SD, mean, mean + SD) as 'high', 'medium', and 'low' confidence keel depths respectively. A lower than average keel:sail ratio would result in a shallower keel draft, so ridges calculated using this ratio that intersect the interpolated

bathymetric line are the most likely to really be grounded ridges. Without validation surveys of the bathymetric depths in the area or the keel:sail ratios in near-shore ice, quantifying the uncertainty in ridge detection is somewhat arbitrary. We feel using the 'high', 'medium', and 'low' designations accounts for the uncertainty in keel depth without artificially quantifying uncertainty when there are too many unknowns.

Figure 10 is updated to include these 'high', 'medium', and 'low' likelihood grounded ridges.

3. Some additional clarification regarding derivation of sail height would be useful
The text states that sail heights are measured relative to the surrounding undeformed ice, the freeboard of which is estimated based on a freezing degree day model. However, there is no specific explanation regarding how the surrounding undeformed ice is defined or identified. Duncan and Farrell refer to height of the local level ice surface ($HL$), which "is computed as the mode of the $hc$ height distribution in 25 km along-track segments", where $hc$ is the height of the sea ice surface above the mean sea surface, corrected for tides and atmospheric conditions. These are important details for interpreting the results presented in this manuscript and I feel they should be included in the text so that the reader does not have to search a separate publication. I would also like to see an explanation of how these 25-km track segments are treated at the coast. Specifically, are they are truncated at the coastline and, if so, what effect might this have on the derivation of level ice height and, therefore, ridge sail heights? Please also see comment 4 about the impact of sea level variations on sail height measurement for grounded ridges.

The processing algorithm presented in Duncan and Farrell excludes ice near shore in order to have the 25-km segments not risk intersecting the coast. For this study, we omit that step of the processing algorithm in order to get the near-shore ice surface heights. The local level ice surface calculated by the the UMD-RDA without that step in near-shore areas do not align with expected ice surface heights, so we do the following:

1.  Find the modal surface height along the ICESat-2 track in the 4km closest to shore. This is to determine the `level ice height` in the region in which most ridging would occur.
2.  Estimate the `thickness of undeformed sea ice` based on the date of observation and a FDD model using available weather station data.
3.  Estimate the `level ice freeboard` using a buoyancy ratio of 917 kg/m3 : 1025 kg/m3. *
4.  Set `sea surface height` to be the height of the freeboard below the modal ice height we calculated in step 1.
5.  The sail height is then the `measured ice surface height` minus the `level ice height` plus the `level ice freeboard`.

We have added an additional figure to the methods section and revised the text accordingly to better illustrate the process we use to extract sail height, float level, and keel depth estimates from the surface elevation measurements.

* This likely slightly underestimates the height of the freeboard (sea ice tends to be slightly lower density than pure ice), which means this algorithm is more likely to slightly undercount grounded ridges, especially in areas with significant snow cover.

4. Some further discussion of local variations in sea level may be required

Although line 129 makes references to a 20-cm tidal range near Utqiagvik and acknowledges this can be a significant fraction of the water depth, I feel further discussion is required. First, I feel the text should recognize that wind-driven variations in sea level are much greater that than the tidal amplitude and can exceed 1 m (e.g. Mahoney et al, 2007b as cited in the manuscript). More significantly however, variations in sea level don't just affect water depth. They can have a much greater effect of the estimation of keel depth due to the way in which sail heights of grounded ridges are measured. Unlike floating ice, the height of grounded ridges relative to the surrounding ice will vary with local sea level and I think this is a more likely explanation than changes in snow depth for the "slight changes in the best estimate keel depths" noted on line 258. For example, the sails of grounded ridges will appear lower when the sea level (and the surrounding non-grounded ice) rises. This would have then have an amplified effect on the estimated keel depth: a 20-cm rise in sea level would reduce the sail heights of grounded ridges by 20 cm, which would then reduce the estimated keel depths by more than 60 cm, depending on the keel/sail ratio used. Hence, the number of grounded ridges could be significantly underestimated if the ridge sails were measured during a surge in local sea level and the authors might consider correcting their sail heights to account for the dynamic topography of the ocean. In effect, the use of the corrected sea ice height, $hc$, to derive the height of level ice, $hL$, (as described by Duncan and Farrell, 2022) may be counterproductive for the purpose of estimating sail height of grounded ridges.

We addressed tidal range because of the possibility of having consistently biased height measurements in a region if ICESat-2 overflight timing lined up with a particularly high or low tide. We are adding discussion of wind-driven changes in local sea level and the impact that would have on the estimates to the manuscript in the Bathymetry section of the Methods, and to our Limitations section.

A wind-driven surge in local sea level that drives the level ice up around a grounded ridge would likely result in this algorithm missing that ridge. With wind-driven variations in sea level height fairly random in time, we do not expect that would cause a systematic bias in under-detecting ridge heights in a particular area. With ground tracks spread across the Alaska Arctic each day for the five month period of this study, there are likely cases where both sea level is elevated compared to a grounded ridge and where it is depressed compared to a grounded ridge.

The idea of correcting sail heights for dynamic topography of the ocean is an interesting one, and one we would like to pursue further in a future study. From the relative surface height measurements alone, it is unclear which are grounded ridges where the surrounding level ice is elevated versus smaller ridges which do not extend deep enough to be grounded. For tracks where we have reliable retrievals of surface height over shore though, it would be possible to account for the dynamic height and more systematically study the impact of short-term sea level variation on ridge grounding. This would be outside the scope of this paper, but certainly an interesting idea to pursue.

5. GEBCO data are of questionable reliability in shallow water

Given the sparsity of sounding points in shallow water and the year-to-year variability of the

bathymetry in these regions, I am somewhat skeptical of the validity of using GEBCO bathymetric data all the way up to the coastline. The GEBCO Type Identifier (TID) Grid, available with GEBCO bathymetry data, show that broad areas of the Alaska coast shallower than ~5 m are coded with the value 41 which indicates the bathymetry values are "Interpolated based on a computer algorithm". I would therefore recommend the authors exercise more caution when identifying grounded features in shallow water. The authors already exclude features in water shallower than the level ice drafter, but I don't understand the rationale for this (see comment 6). Instead, I recommend the authors establish a shallow-water cut-off value, based on the GEBCO TID Grid. This value might vary regionally according to the density of sounding points used in the GEBCO grid, but this would provide a data-driven rationale for excluding ridges in shallow water. Also, under the topic of bathymetric uncertainty, I don't understand how the orange areas in Figures 4-7 were derived. Line 209 states "The orange shading represents positional uncertainty in the bathymetry line", but the width of the orange shading would suggest that the positional uncertainty is on the order of hundreds of meters, which seems too high.

In the 2024 GEBCO product, the TID Grid does not identify shallow water areas as consistently interpolated (Type 41). We have included in this response a map of type identifiers both across northern Alaska and in the area around Utqiagvik and Elson Lagoon (following page). There are singlebeam depth measurements (Type 11) in shallow water ($\leq$ 2 m) including in lagoon areas. GEBCO does not establish a shallow-water cutoff.

That said, there certainly are issues with depth estimates at shallow water depths. In re-running the analysis we implement a general shallow water cutoff at 3m depth. This is in part due to the uncertainty in shallow water measurements, but generally to exclude features inside the lagoon areas. The orange bars in the original version of figures 4-7 reflected the 500m spatial resolution of the GEBCO product. We have removed these and instead address issues regarding the coarse spatial resolution of the GEBCO product in the discussion.

**GEBCO Type ID Map**

[Figure]

0 Land
11 Singlebeam - depth value collected by a single beam echo-sounder
41 Interpolated based on a computer algorithm - depth value is an interpolated value based on a computer algorithm (e.g. Generic Mapping Tools)
71 Unknown source - depth value from an unknown source

6. Exclusion of ridges in water shallower than draft of undeformed ice seems unnecessary

I don't understand the rationale for excluding ridges in water shallower than the draft of undeformed ice. For such ridges to exist at the location, they must have become grounded at some stage when the ice was thinner. Hence, excluding ridges based in undeformed ice draft creates the possibility of a scenario in which a ridge is counted as grounded on one day, but excluded the next. I would therefore discourage the authors from excluding features on this basis and instead simply apply a shallow water cut-off as explained in comment 5.

We implement a shallow water cutoff of 3m to distinguish grounded ridges from bottom-fast ice along the coastline. Without a shallow water cutoff in the GEBCO product to use, we found that 3m excluded features inside lagoon areas and rubble on beaches without excluding too many grounded ridges. Using 5m for the shallow water cutoff excluded a lot more high-likelihood grounded ridges and seemed too high an arbitrary threshold given the data available.

7. Some more careful reading of Mahoney et al (2007a, as cited in the text) may be required
The text makes many references to the work of Mahoney et al (2007a), which I feel mischaracterize what is written in the cited work. For example, on line 308, the text states "Mahoney et al. (2007a) assumes a sail width of 100 m for grounded ridges", but this is a rather inaccurate description of what was written in that paper. For the purposes of estimating the possible spatial density of grounded ridges, Mahoney et al assumed that the keels of ridges deep enough to become grounded were on the order of 100m wide. This is not the same as assuming grounded ridge sails are 100 m wide.

We thank the reviewer for clarification on Mahoney et al.'s results. We removed the reference to Mahoney et al. (2007a) from this paragraph so as to not misrepresent that work and instead reference the ridge width statistics presented in SKS 2012.

Similarly, line 315 begins with the statement "Mahoney et al. (2007a) suggested grounded ridges discontinuously pin the landfast ice edge, roughly every 30 km". In this case, I feel the authors are taking what is written in the cited article somewhat out of context. Instead, Mahoney et al. wrote that grounded ridges would be spaced approximately every 30 km if the thickness distribution of landfast ice was the same as that measured offshore in the drifting pack ice. And they go on to note that grounded ridges are observed more closely spaced than this, suggesting that they are produced through "in-situ" grounding process (i.e. like those illustrated in Figure 1 of this manuscript).

Thank you for this clarification, we have revised the discussion around the frequency of grounded ridges observed in the tracks:

Our results suggest, on average, 1.9 and 2.5 medium-confidence grounded ridges per ICESat-2 ground track in the Chukchi and Beaufort regions respectively (with ranges of 0.4 - 5.1 and 0.5 - 6 .7 between high and low confidence ridges in the two basins). Given these tracks pass over an effectively random sampling of coastal Arctic sea ice, and that there are on average >1 ridge per track, we can infer that any along-shore section of ice is likely grounded by at least one ridge, though there are certainly some places without any grounded ridges (approximately 16% of ICESat-2 surface height profiles examined contained no grounded ridges at any confidence level, and 40% had no high-confidence grounded ridges). We also find that the average high confidence grounded ridge location was 23 % and 17% of the distance between the coastline and landfast ice edge in the Chukchi and Beaufort Sea regions, respectively. Together, this suggests that coastal sea ice is regularly supported by frequent grounded ridges in the shallow water (< 10 m depth) and infrequent grounded ridges in the deeper stamuki zone. This is a higher spatial density in the shallow water zone than in pack ice (Mahoney et al 2007a), which would suggest a combination of in situ ridge formation and advected ridges getting stuck in shallow waters.

Lastly, on line 320, the text claims "Mahoney et al. (2007a) assumes that these grounded ridge features are located close to the SLIE", but I again feel this does not accurately represent what Mahoney et al. wrote. Rather than making any general assumption that grounded ridges are located close to the SLIE, they identify "nodes" where the SLIE occurs most commonly and suggest that these correspond to the location of interannually recurring grounded features. Additionally, I recommend the authors consider citing Mahoney et al's more recent 2014 paper, which expands on the discussion of these nodes and provides photographic documentation of a grounded ridge in the location of a node that happens to be very close to the location of line C in Figure 2. I also feel that some further discussion of the location of the nodes identified by Mahoney et al in relation to the distribution of grounded features shown in Figure 10 would be a valuable addition to the text.

We have added further reference to Mahoney et al.,'s 2014 discussion of nodes in the landfast ice in the discussion. Thank you for sharing the shapefile for the locations of nodes identified in Mahoney et al. 2014 – we have added these to the updated version of Figure 10. Figure 10 will be further revised to include three subsets with zoomed-in views of areas around three of the nodes (near longitudes -162, -155, and - 148). The text will be revised to include this additional discussion:

Nodes from Mahoney et al., 2014 are included in the map on Figure 10, representing locations in which the landfast ice edge is statistically more likely to occur. Within each location, we find grounded ridges detected from our algorithm. In some nodes (e.g., the node at -162 longitude, closest to Point Lay, inset A), we see a cluster of grounded ridges, including high-confidence grounded ridges, at the landfast ice edge, pinning that edge in place but not providing any support for an extended SLIE. In other nodes (e.g., the node at -155 longitude, close to Point Barrow, inset B), relatively few grounded ridges are close to the landfast ice edge, and they are all low-confidence detections. This suggests that the local ice dynamics support a more extensive SLIE. The node at -148 longitude (near Prudhoe Bay, inset C) shows a combination of these behaviors with many grounded ridges close to the ice edge at the west end, but a larger SLIE at the east end.

**Minor comments**

Line 9 (and throughout the text): The word "Alaskan" is a noun referring to someone from Alaska. When used an adjective, the correct term is Alaska.

This will be corrected throughout the text.

Figure 1: This figure illustrates one of two ways in which a sea ice ridge can become grounded. The other way involves advection of a deep-keeled ridge into shallow water. Mahoney et al (2007b as cited in the manuscript) speculate that the latter is more likely to create a gouge in the seafloor and is therefore more significant for stabilizing the landfast ice. The authors may wish to acknowledge both ways of creating a grounded ridge and the differences between them.

We acknowledge the advection of deep-keeled ridges in the text, but agree that this can be more explicit and have updated the introduction accordingly. We have added an alternate development track in Figure 1 to make this mechanism more clear, and we have added discussion throughout the manuscript to better reflect this mechanism.

Figure 2: Why did the authors select these ICESat-2 tracks? Had they selected tracks that intersected the coast closer to Utqiagvik, they could fallen within the footprint of UAF's sea ice radar and there's a good chance that they would have intersected whaling trails where sea ice thickness has been routinely measured since 2007 (https://arctic-aok.org/data-sources/whaling-trail-mapping/). These data are both reference could have provided useful validation for the thickness of level ice in the region. Also, why is Line C truncated before the 20m isobath? From several years of observations, the landfast ice in this region is commonly anchored by grounded ridges just beyond the end of Line C.

These tracks were picked for showing a variety of sample ridge conditions, as part of the initial case study. Cloud cover/precipitation can prevent surface height retrievals, and these three tracks had very few segments of missing data in the regions around Utqiagvik for ground tracks both in January and April of that year. There are not available tracks with full coverage in both January and April that are closer to the Utqiagvik sea ice radar during the 2021-2022 winter. We do have a pair of tracks from December and March, but the December track is prior to the formation of grounded ridges.

There are a few single tracks (without repeat) that intersect the sea ice radar area. Grounded ridges detected in the sea ice radar area are approximately 1.2km from shore: this is consistent with the persistent features in the sea ice radar. It is harder to see those features in individual images in the sea ice radar (more apparent in the video animation), so we are not going to put an ice radar image in the paper, but we have added discussion to section 4.1 (Detection of Grounded ridge features) describing this additional means of validation.

We have updated the figure for Line C beyond the 20m isobath. In the January track, there are no further grounded ridges beyond what was shown in the prior version. In April, there is one ridge at 14km from shore that barely intersects the bathymetry line for the largest keel:sail ratio estimate ('low' confidence ridge) at 16m bathymetric depth. We have updated the discussion accordingly.

The whaling trail mapping effort has a lot of potential for a larger validation study of this approach: while we were only able to process a single year of the ICESat-2 data for this project, we hope to secure funding for a larger effort covering more years and more area. Working with the local community for detailed sail height measurements (and any keel geometry information) would be an important part of that.

Lines 265-267: There are multiple assertions here that should be supported by references. I think I know what the authors mean by the "classic" ridge, but I have a suspicion that its classic status derives from simplifications adopted in many illustrations of coastal ice over the years together with a bias in the early literature toward ridges in the Beaufort Sea. Also, can the authors provide a reference supporting the "prevalance of shear" on the Chukchi Side of Point

Barrow? The climatological prevailing wind is from the east, with creates a lot more divergence than shear and a lot less shear than on the Beaufort side of Pot Barrow

We reviewed the sea ice radar animations for the 2021-2022 winter: while there isn't overlap with the ice radar field of view and the ICESat-2 track, the radar does indicate the direction of pack ice movement during the periods of interest. We have changed the description in this section to be more specific about the dynamics at work here:

The spatial resolution of the SAR imagery poses a challenge for unambiguous interpretation of grounded ridges and is an insufficient tool by itself for determining where exactly ridge features form or for characterizing their geometry. The nearby Utqiagvik Sea Ice Radar (UAF, 2022a) shows pack ice motion towards the east/north-east, resulting in a compressive ridge building event on February 22, 2022. This ice motion, combined with the existing land-fast ice in the area, creates ridges that are not necessarily parallel to shore. We measure ridge width using the distance from shore, so a ridge that is not parallel to shore will overestimate the ridge width. While the field of view of the radar does not overlap with this track, the orientation of the bright patches in the SAR imagery near ridge 3 (forming between Feb 18 and Mar 2) at an angle relative to shore is consistent with the ridges visible in the Utqiagvik sea ice radar from that period.

Line 337-338: The reasoning behind this explanation in the final sentence of this paragraph is not clear to me. If grounded ridges in the same water depth are higher in the Beaufort Sea than in the Chukchi, this means the ice is piled higher above the waterline in the Beaufort. I can envision a few mechanisms that might cause such a difference (for example in-situ grounding vs advection of deep-keeled ridges in shallow water) and how they might relate to coastal aspect or parent ice thickness, but I feel most readers would benefit from further explanation.

We appreciate this feedback and have added additional explanation to the text:

In the Beaufort Sea, grounded ridges at the same bathymetric depth tend to be slightly taller than those in the Chukchi region. This difference in ridge height may be attributed to a combination of factors, including shoreline orientation relative to predominant ice drift, the thickness of the parent ice floe, and the processes by which grounded ridges form. Drift patterns in the Beaufort region bring thicker ice closer to the shoreline on a trajectory roughly perpendicular to shore. This increases the likelihood of advection of deep-keeled ridges from the pack ice into shallow waters.

---

## Author Comment (AC2)

Review of Grounded ridge detection and characterization along the Alaskan Arctic coastline using ICESat-2 surface height retrievals by Lange et all.

Responses in blue text.

In this paper, the authors introduce a novel method for detecting grounded sea ice ridges. They compare characteristics of grounded ridges in the Chukchi Sea with the ones of grounded ridges in the Beaufort Sea. They show that many ridges are grounded in water significantly shallower that the traditional stamuki zone.

I find this novel approach very interesting. As stated by the authors, this new method could have many useful applications. Overall, I find that the paper is well written. At a few places, however, I would like the authors to clarify the text and to use more appropriate terms. For example, ridge depth is confusing while sail height and keel depth are more clear. Also, I think that often they just use the word 'ice' when they should use the expression 'level ice'.

Below the authors will find a few more major comments and a list of minor comments.

I recommend major revisions (to give them more time) but the authors should really see it as moderate revisions.

We thank the reviewer for their kind words and helpful comments. Individual comments are addressed below:

MAJOR COMMENTS

1) I think Fig.8 should be improved. The panels should show PDFs instead of histograms. It would then be easier to compare the distributions of grounded ridges in the Chukchi Sea to the ones in the Beaufort Sea.

Figure 8 will be revised to reflect your suggestions: we show PDFs instead of histograms. Consistent with other reviewer recommendations, we have also added background lines to indicate the pdfs for low and medium-confidence grounded ridges in addition to the high-confidence ridges.

2) The authors discuss the limitations of their approach in section 4.5 and acknowledge the uncertainty associated with their bathymetry data. First of all, I think there should be some clarifications on how they estimate the uncertainty in the bathymetry data. I don't think this

is mentioned (Fig.5-7). Given their low resolution bathymetry data, I think a possible improvement to their detection algorithm would be to consider a distribution of bathymetry at the location of possible grounded ridges. The mean of the distribution could come from the low resolution GEBCO data and the distribution around the mean could be estimated from the NCEI high resolution data. The algorithm could then estimate the probability of contact between the keels and the sea floor. I know this is beyond the scope of this paper but I think this should be discussed and presented as a possible improvement to the existing algorithm. I think the authors should have a look at this paper which introduces this idea for a model grounding parameterization:

Dupont et al., A probabilistic seabed–ice keel interaction model, the Cryosphere, 2022.

Thank you for this suggestion! While a probabilistic model of seabed interaction is outside the scope of this paper, we will consider it in future work based on the approach described here. In the updated manuscript, we have added discussion acknowledging the potential of this approach.

We have removed the bands in Figure 5-7 indicating the width of the pixels in the GEBCO data (consistent with other reviewer comments) but added discussion in this section in order to address the position uncertainty associated with the coarse product. The updated approach considers a range of possible keel depths associated with each ridge: we have edited the language throughout the manuscript to better reflect the uncertainty in both keel depth estimate and bathymetric depth.

3) I really like Fig.10. The fact that there is landfast ice seaward of the "last" grounded ridges is an indication that sea ice has some tensile strength (or cohesion). I see an interesting study that could be conducted: the data of grounded ridges along with maps of landfast ice (the blue shading) could be used to estimate the tensile strength of sea ice. Again this is beyond the scope but could be discussed by the authors if they want to. The authors could have a look at this paper which describes how the viscous-plastic sea ice rheology can be modified in order to add tensile strength:

Konig and Holland, Modeling Landfast Sea Ice by Adding Tensile Strength, Journal of Oceanography, 2010.

This is also a very helpful comment for future work – we will address this idea in the conclusions of the paper.

MINOR COMMENTS

The minor comments will be addressed individually in the paper, most of which are small changes in wording and phrasing. We appreciate your attention to detail on these comments – they certainly make it a better paper! We will make the recommended changes for language/grammar/minor organization. More substantive minor comments are addressed individually below.

1) Abstract line 3: remove 'to describe an approach'.

2) line 27: It is not clear what you mean by 'unstable ice conditions' and the link with later freeze-ups and earlier thaws.

We have re-written this whole paragraph to make it more clear (consistent with other reviewer comments):

"An increase in ice breakout events and shorter shorefast ice seasons during recent years endangers hunters whose safety depends on knowledge of ice dynamics (Gearheard et al., 2006; Mahoney et al., 2014; George et al., 2004). Since 1980, the volume of ice in the Arctic Ocean is thought to have declined by 75% (Overland et al., 2014), and observations show that later ice freeze-ups and earlier thaws are occurring throughout the Arctic as a result of shortening winters (Mahoney et al., 2014; Stammerjohn et al., 2012) and thinning ice (Kwok et al., 2009; Rothrock et al., 1999; Laxon et al., 2013; Mahoney et al., 2009; Howell et al., 2016; Gerland et al.,2008). Mahoney et al. (2014) show that dates of first ice, break-up and ice-free conditions for landfast ice in the Beaufort and Chukchi Seas are changing up to 1 week/decade with later formation and earlier breakup dates each year, excluding break-up in the Beaufort which shows no conclusive trend."

3) lines 30-38: This paragraph needs to be reworked. After the reference to Eicken et al. the first sentence should be 'Grounded ridges form either when a compression ridge drifts into shallower waters and gets stuck in the sea floor or from an in situ collision between the shorefast ice and drifting pack ice'. Then the next sentence could start by 'Figure 1A describes how grounded ridges are created from the collision of the drifting sea ice with the shorefast ice'...

4) line 40: replace 'for the ice to reach' by 'for the keel to reach'.

5) line 40: 'the thickness of the ice'...do you mean 'the thickness of the level ice'? Please clarify.

6) line 44: 'Assuming some typical sea ice thickness...' are you referring to level ice again?

7) line 56: Shouldn't you remove 'spring' in 'spring shore-fast ice season'?

8) line 64: replace 'is likely to be grounded in the sea floor' by 'is likely to include grounded ridges as anchoring points'.

9) line 97: remove one 'using'.

10) line 115: replace 'therefor' by 'therefore'.

11) Title for section 3.1: Should it be: 'Freezing degree day model to estimate level ice thickness'?

12) Section 3.1: What about snow in your freezing degree day model?

13) line 190: Are you sure about citing  Yu et al. 2014? I don't remember that they discuss the ratio of sail height to keel depth (and that it varies with the geographical location).

Good catch, this should have been Strub-Klein and Sudom, 2012. We have fixed this in the manuscript.

14) lines 212-214: rephrase. Just say that the bathymetry is interpolated to the center (?) of the keel or something like that.

We have revised the test to clarify that the bathymetric depth recorded for each ridge is the interpolated bathymetric depth at the location of the maximum sail height.

"We then record several characteristics of the identified grounded ridges. The right panel of Figure 5 shows how we characterize each grounded ridge feature: the start and end of the ridge in the horizontal direction is used to calculate width, height of the sail, and depth of the bathymetric contour where it intersects the keel depth estimate. Bathymetric depth is interpolated to the center of the predicted keel location."

15) line 213: replace 'defined' by 'define'.

16) line 233: replace 'suggest' by 'suggests'.

17) line 335: You mean 'keel depth' instead of ridge depth? By the way you should change the labels in Fig.8 and 10. It should be sail height and keel depth.

We have updated the labels to sail height and keel depth.

18) line 350: rephrase

19) line 352: you should add that some of these ridges could drift in these shallow areas and get grounded.

Yes! We will specify that it is possible for a drifting floe containing a ridge to ground near shore and get stuck, especially if there are fluctuations in water level related to wind direction. We've added to Figure 1 to better reflect this mechanism.

20) Fig.9: Would it be good to have a second column showing the maximum instead of just the mean?

Great suggestion. We have updated the figure to include maximums and means for each of the ridge geometry parameters.

21) line 423: '...seasonal landfast ice variability...' do you mean '...interannual landfast ice variability...'?

22) lines 427-428: Should you move up this sentence where you discuss models (line 423 for example)? It should also be rephrased. I am not sure I understand what you mean.

We have rephrased this and rearranged the discussion of grounded ridges in models to be more clear:

"A wider application of this approach to shorefast ice observation could be used to validate modeling efforts at capturing landfast ice dynamics (e.g., (Lemieux et al., 2015)).Grounded ridge formation occurs at sub-grid scale in current sea ice models, though these ridges impact larger scale ice movement and near-shore dynamics. Modern models are incapable of reflecting interannual landfast ice variability which impacts sea ice thickness and concentration results, halocline stability (Itkin et al., 2015), upwelling estimations (Kasper and Weingartner, 2015), and brine expulsion (Selyuzhenok et al., 2015). Shorefast ice also blocks momentum flux between the atmosphere and ocean, isolating the ocean from wind-driven mixing and limiting upwelling, and causes river plumes to extend to further distances under the ice (Granskog et al., 2005). Ultimately, the approach outlined in this study creates opportunities to observe grounded ridge properties across much larger spatial scales, opening potential avenues for improved understanding of coastal Arctic processes."

Jean-Francois Lemieux

---

## Author Response (AR1)

Line by line responses to the review by Jean-Francois Lemieux:

Review of Grounded ridge detection and characterization along the Alaskan Arctic coastline using ICESat-2 surface height retrievals by Lange et all.

We appreciate your attention to detail in the review of this paper! General responses are in blue text below, specific line-by-line comments are in red.

In this paper, the authors introduce a novel method for detecting grounded sea ice ridges. They compare characteristics of grounded ridges in the Chukchi Sea with the ones of grounded ridges in the Beaufort Sea. They show that many ridges are grounded in water significantly shallower that the traditional stamuki zone.

I find this novel approach very interesting. As stated by the authors, this new method could have many useful applications. Overall, I find that the paper is well written. At a few places, however, I would like the authors to clarify the text and to use more appropriate terms. For example, ridge depth is confusing while sail height and keel depth are more clear. Also, I think that often they just use the word 'ice' when they should use the expression 'level ice'.

Below the authors will find a few more major comments and a list of minor comments.

I recommend major revisions (to give them more time) but the authors should really see it as moderate revisions.

We thank the reviewer for their kind words and helpful comments. Individual comments are addressed below:

MAJOR COMMENTS

1) I think Fig.8 should be improved. The panels should show PDFs instead of histograms. It would then be easier to compare the distributions of grounded ridges in the Chukchi Sea to the ones in the Beaufort Sea.

Figure 8 (now Figure 9) has been revised to reflect your suggestions: we show PDFs instead of histograms. Consistent with other reviewer recommendations, we have also added background lines to indicate the pdfs for low and medium-confidence grounded ridges in addition to the high-confidence ridges.

2) The authors discuss the limitations of their approach in section 4.5 and acknowledge the uncertainty associated with their bathymetry data. First of all, I think there should be some clarifications on how they estimate the uncertainty in the bathymetry data. I don't think this is mentioned (Fig.5-7).  Given their low resolution bathymetry data, I think a possible improvement to their detection algorithm would be to consider a distribution of bathymetry at the location of possible grounded ridges. The mean of the distribution could come from the low resolution GEBCO data and the distribution around the mean could be estimated from the NCEI high resolution data. The algorithm could then estimate the probability of

contact between the keels and the sea floor. I know this is beyond the scope of this paper but I think this should be discussed and presented as a possible improvement to the existing algorithm. I think the authors should have a look at this paper which introduces this idea for a model grounding parameterization:

Dupont et al., A probabilistic seabed–ice keel interaction model, the Cryosphere, 2022.

Thank you for this suggestion! While a probabilistic model of seabed interaction is outside the scope of this paper, we will consider it in future work based on the approach described here. In the updated manuscript, we have added discussion acknowledging the potential of this approach.

We have removed the bands in Figure 5-7 (now 6-8) indicating the width of the pixels in the GEBCO data (consistent with other reviewer comments) but added discussion in this section in order to address the position uncertainty associated with the coarse product. The updated approach considers a range of possible keel depths associated with each ridge: we have edited the language throughout the manuscript to better reflect the uncertainty in both keel depth estimate and bathymetric depth.

3) I really like Fig.10. The fact that there is landfast ice seaward of the "last" grounded ridges is an indication that sea ice has some tensile strength (or cohesion). I see an interesting study that could be conducted: the data of grounded ridges along with maps of landfast ice (the blue shading) could be used to estimate the tensile strength of sea ice. Again this is beyond the scope but could be discussed by the authors if they want to. The authors could have a look at this paper which describes how the viscous-plastic sea ice rheology can be modified in order to add tensile strength:

Konig and Holland, Modeling Landfast Sea Ice by Adding Tensile Strength, Journal of Oceanography, 2010.

This is also a very helpful comment for future work – we briefly address this idea in the conclusions of the paper (line 525).

MINOR COMMENTS

We appreciate your attention to detail on these comments – they certainly make it a better paper!

1) Abstract line 3: remove 'to describe an approach'.

We removed that part of line 3.

2) line 27: It is not clear what you mean by 'unstable ice conditions' and the link with later freeze-ups and earlier thaws.

We have re-written this whole paragraph to make it more clear (consistent with other reviewer comments), starting on line 29.

3) lines 30-38: This paragraph needs to be reworked. After the reference to Eicken et al. the first sentence should be 'Grounded ridges form either when a compression ridge drifts into shallower waters and gets stuck in the sea floor or from an in situ collision between the shorefast ice and drifting pack ice'. Then the next sentence could start by 'Figure 1A describes how grounded ridges are created from the collision of the drifting sea ice with the shorefast ice'...

This paragraph has been re-written, starting on line 34.

4) line 40: replace 'for the ice to reach' by 'for the keel to reach'.

This sentence has been rephrased on line 49.

5) line 40: 'the thickness of the ice'...do you mean 'the thickness of the level ice'? Please clarify.

This entire sentence has been rephrased for additional clarification, line 49.

6) line 44: 'Assuming some typical sea ice thickness...' are you referring to level ice again?

Yes, this is again referring to level ice. We rephrased the sentence on line 54.

7) line 56: Shouldn't you remove 'spring' in 'spring shore-fast ice season'?

We removed 'spring' from this sentence. We also moved this paragraph earlier in the introduction for better flow. It snow starts on line 26.

8) line 64: replace 'is likely to be grounded in the sea floor' by 'is likely to include grounded ridges as anchoring points'.

We have made this change on line 74.

9) line 97: remove one 'using'.

We have revised this sentence on line 95.

10) line 115: replace 'therefor' by 'therefore'.

This typo has been corrected on line 131.

11) Title for section 3.1: Should it be: 'Freezing degree day model to estimate level ice thickness'?

We have retitled the section (line 137).

12) Section 3.1: What about snow in your freezing degree day model?

These freezing degree day models are empirical relationships derived from snow-covered ice: they do not account for actual precipitation measurements.

13) line 190: Are you sure about citing Yu et al. 2014? I don't remember that they discuss the ratio of sail height to keel depth (and that it varies with the geographical location).

Good catch, this should have been Strub-Klein and Sudom, 2012. We have fixed this in the manuscript on line 201.

14) lines 212-214: rephrase. Just say that the bathymetry is interpolated to the center (?) of the keel or something like that.

We have revised the text starting in line 238 to clarify that the bathymetric depth recorded for each ridge is the interpolated bathymetric depth at the location of the maximum sail height. We have also added some markup to figure 5 to better show how the bathymetric depth of a ridge is selected.

15) line 213: replace 'defined' by 'define'.

We have edited this sentence, now in line 240.

16) line 233: replace 'suggest' by 'suggests'.

We have made this change in line 296.

17) line 335: You mean 'keel depth' instead of ridge depth? By the way you should change the labels in Fig.8 and 10. It should be sail height and keel depth.

We have clarified in the text and the figure captions that we mean the bathymetric depth at the location of the maximum height on the sail. We have updated the labels in figure 8 (now figure 9) and figure 10 (now 11) to sail height and bathymetric depth.

18) line 350: rephrase

We have rephrased this entire paragraph, starting at line 380.

19) line 352: you should add that some of these ridges could drift in these shallow areas and get grounded.

We added language to clarify the mechanism of a drifting floe containing a ridge to ground near shore and get stuck, especially if there are fluctuations in water level related to wind direction. We've added to Figure 1 to better reflect this mechanism, along with adding text in the introduction (starting in line 34) and discussion starting in line 388.

20) Fig.9: Would it be good to have a second column showing the maximum instead of just the mean?

Figure 10 (formerly figure 9) has been updated: we looked at the maximum, but what ended up making more sense was to include both the mean and the furthest grounded ridge from shore.

21) line 423: '...seasonal landfast ice variability...' do you mean '...interannual landfast ice variability...'?

Yes, this has been corrected in line 521 (there has been some reorganization).

22) lines 427-428: Should you move up this sentence where you discuss models (line 423 for example)? It should also be rephrased. I am not sure I understand what you mean.

We have rephrased this and rearranged the discussion of grounded ridges in models to be more clear. This sentence is now in the paragraph starting in line 518.

Thank you again for your helpful comments!

Line-by-line responses to Andrew Mahoney:

Grounded ridge detection and characterization along the Alaska Arctic coastline using ICESat-2 surface height retrievals
by Kennedy A. Lange, Alice C. Bradley, Kyle Duncan, and Sinéad L. Farrell
Submitted to The Cryosphere Discussions
https://doi.org/10.5194/egusphere-2024-1885
Review Aug 30, 2024

Thank you for the thoughtful and detailed review! We have included general responses in blue text, and more specific line-by-line comments in red.

**Summary**
This manuscript presents an analysis of the distribution of grounded sea ridges near the Alaska coast in the Chukchi and Beaufort Seas based on a combination of ridge sail heights derived from ICESat-2 altimetry data and GEBCO bathymetry data. The approach can be boiled down to 3 key steps. First, the individual ridges are identified and their sail heights determined using ICESat-2/ATLAS geolocated photon height (ATL03) data following the University of Maryland Ridge Detection Algorithm (Duncan and Farrell, 2022). Second, the range of likely keel depths associated with each ridge sail is estimated using sail height/deel depth ratios derived data reported by Strub-Klein and Sudom (2012). Lastly, grounded ridges are identified as those with keel depths exceeding the water depth at their location. The result of this approach is an impressively high fidelity picture of grounding locations within the landfast ice during the 2021-22 winter that suggests grounded ridges for in deeper water in the Beaufort Sea than the Chukchi, but overall the vast majority of grounded ridges are found in waters less than 15 m deep.

Overall, I enjoyed reading this paper and I believe the work will make a valuable contribution to our understanding of landfast stability and sea ice / seafloor interaction. However, in preparing this review I identified several concerns relating to a lack of methodological detail, missing discussion of relevant physical processes, under-utilization of uncertainty calculations, and perhaps an incomplete reading of some of the cited literature. Being quite familiar with landfast ice in this region, I find the sparsity of grounded ridges near the 20-m isobath surprising, particularly in the Beaufort Sea, and in its current form, I feel the manuscript leaves me with too many questions to take such surprising results at face Value.

I want to stress that I would very much like to see this work published. I don't think any of my concerns should be too difficult to address, and once they are, I would be feel more comfortable about accepting the authors finding regarding the water depth distribution of grounded ridges. I have provided detailed comments below explaining my concerns and I have attempted to make constructive recommendations for improving the manuscript.

We thank the reviewer for their kind words and helpful comments which we address individually below in blue text:

**Major Comments**

1. Why are the keel/sail ratios different from those reported by Strub-Klein and Sudom?
The principle findings of this manuscript critically depend on the ratio between sail height and keel depth. I am therefore surprised to see no discussion regarding the difference between the ratios determined via the linear regressions illustrated in Figure 3 and those reported by Strub-Klein and Sudom. For the

Chukchi Sea, the authors find a linear regression with a slope of 3.37 and confidence intervals with slopes of 2.92 and 3.82, while Strub-Klein and Sudom report a mean ratio of 3.92. In the Beaufort Sea, the difference is greater with Fig 3 showing a regression slope of 3.49 and confidence interval slopes of 2.74 and 4.25 while Strub-Klein and Sudom report a mean value of 4.72. Given the significance of these ratios in the determining the apparent presence or absence of grounded features, I feel these differences should be discussed in some depth. For example, how many more grounded features would be identified beyond the 10 m isobath if the authors used Strub-Klein and Sudom's Ratios?

We had used a first-order polynomial fit to the sail heights/keel depths in order to account for the thickness of undeformed ice: using a consistent ratio can introduce errors in smaller ridges. That said, those small ridges are (a) likely to be filtered out given the requirement to exceed 0.7m height above modal surface level that comes from the UMD-RDA, and (b) unlikely to be grounded ridges anyways. We have re-run the analysis, using the mean ratios identified in Strub-Klein and Sudom ratios and accounted for the variability in keel/sail ratios in the observations by using a full standard deviation for the range of values. Our new ratios reflect the 3.92 and 4.72 sail:keel ratios for Chukchi and Beaufort seas, with standard deviations of 1.99 and 1.78 respectively. We treat the three ratios for each basin (mean - SD, mean, mean + SD) as 'high', 'medium', and 'low' confidence keel depths respectively.

All of the figures (5-11) throughout the paper have been updated based on these updated keel depth ratios. Major changes to text related to this are in section 3.2-3.3 and throughout the results section.

2. Uncertainties in keel depth and water depth are discussed but under-utilized
The authors list a number of sources of uncertainty that could affect their findings and on lines 131-132, the text states "These uncertainties are acknowledged in this study and the results are interpreted accordingly". I commend the authors for including confidence intervals on many of their figures, but the only subsequent reference to them is on lines 382-383 where the text reads "For ridges where the whole 95% confidence interval (indicated by the purple shading) intersects the sea floor, and the features remain persistent between two dates, we can detect grounded ridges with some certainty". Aside from this brief and rather qualitative statement, I can find no other indication that the confidence intervals are taken into account when identifying grounded ridges. As a result, the principal way in which the authors account for uncertainties in keel depth and bathymetry appears to be simply acknowledging their existence and the occasional use of qualifying terms like "potential" and "possible" when referring to the identification of grounded ridges. Instead, I encourage the authors to consider assigning some quantitative level uncertainty to each grounded feature based on the degree of overlap between the confidence intervals for keel depth and bathymetry. At the least, the authors need to clarify whether or not the confidence intervals play any role in the identification of the grounded ridges illustrated in Figure 10.

Thank you for this feedback: we have re-run the analysis detecting grounded ridges from keel depth estimates as described above. We use the three ratios for each basin (mean - SD, mean, mean + SD) as 'high', 'medium', and 'low' confidence keel depths respectively. A lower than average keel:sail ratio would result in a shallower keel draft, so ridges calculated using this ratio that intersect the interpolated bathymetric line are the most likely to really be grounded ridges. Without validation surveys of the bathymetric depths in the area or the keel:sail ratios in near-shore ice, quantifying the uncertainty in ridge detection is somewhat arbitrary. We feel using the 'high', 'medium', and 'low' designations accounts for the uncertainty in keel depth without artificially quantifying uncertainty when there are too many unknowns.

Each of the figures 5-8 are updated to show the 'high', 'medium' and 'low' confidence keel depths. Table 1 now shows the ridge statistics for each of the three categories. Figures 9-10 now have all confidence levels of ridges included. Figure 11 (formerly figure 10) is updated to include these 'high', 'medium',

and 'low' likelihood grounded ridges, color-coded. The discussion of the ridge detection approach (section 3.3, page 6 line 54 - page 7 line 32) explains the differences, and the Results section has been updated throughout accordingly.

3. Some additional clarification regarding derivation of sail height would be useful
The text states that sail heights are measured relative to the surrounding undeformed ice, the freeboard of which is estimated based on a freezing degree day model. However, there is no specific explanation regarding how the surrounding undeformed ice is defined or identified. Duncan and Farrell refer to height of the local level ice surface ($HL$), which "is computed as the mode of the $hc$ height distribution in 25 km along-track segments", where $hc$ is the height of the sea ice surface above the mean sea surface, corrected for tides and atmospheric conditions. These are important details for interpreting the results presented in this manuscript and I feel they should be included in the text so that the reader does not have to search a separate publication. I would also like to see an explanation of how these 25-km track segments are treated at the coast. Specifically, are they are truncated at the coastline and, if so, what effect might this have on the derivation of level ice height and, therefore, ridge sail heights? Please also see comment 4 about the impact of sea level variations on sail height measurement for grounded ridges.

The processing algorithm presented in Duncan and Farrell excludes ice near shore in order to have the 25-km segments not risk intersecting the coast. For this study, we omit that step of the processing algorithm in order to get the near-shore ice surface heights. The local level ice surface calculated by the the UMD-RDA without that step in near-shore areas do not align with expected ice surface heights, so we do the following:

1. Find the modal surface height along the ICESat-2 track in the 4km closest to shore. This is to determine the `level ice height` in the region in which most ridging would occur.
2. Estimate the `thickness of undeformed sea ice` based on the date of observation and a FDD model using available weather station data.
3. Estimate the `level ice freeboard` using a buoyancy ratio of 917 kg/m3 : 1025 kg/m3. *
4. Set `sea surface height` to be the height of the freeboard below the modal ice height we calculated in step 1.
5. The sail height is then the `measured ice surface height` minus the `level ice height` plus the `level ice freeboard`.

We have added an additional figure to the methods section (figure 3) and revised the text accordingly to better illustrate the process we use to extract sail height, float level, and keel depth estimates from the surface elevation measurements.

* This likely slightly underestimates the height of the freeboard (sea ice tends to be slightly lower density than pure ice), which means this algorithm is more likely to slightly undercount grounded ridges, especially in areas with significant snow cover.

4. Some further discussion of local variations in sea level may be required
Although line 129 makes references to a 20-cm tidal range near Utqiagvik and acknowledges this can be a significant fraction of the water depth, I feel further discussion is required. First, I feel the text should recognize that wind-driven variations in sea level are much greater that than the tidal amplitude and can exceed 1 m (e.g. Mahoney et al, 2007b as cited in the manuscript). More significantly however, variations in sea level don't just affect water depth. They can have a much greater effect of the

estimation of keel depth due to the way in which sail heights of grounded ridges are measured. Unlike floating ice, the height of grounded ridges relative to the surrounding ice will vary with local sea level and I think this is a more likely explanation than changes in snow depth for the "slight changes in the best estimate keel depths" noted on line 258. For example, the sails of grounded ridges will appear lower when the sea level (and the surrounding non-grounded ice) rises. This would have then have an amplified effect on the estimated keel depth: a 20-cm rise in sea level would reduce the sail heights of grounded ridges by 20 cm, which would then reduce the estimated keel depths by more than 60 cm, depending on the keel/sail ratio used. Hence, the number of grounded ridges could be significantly underestimated if the ridge sails were measured during a surge in local sea level and the authors might consider correcting their sail heights to account for the dynamic topography of the ocean. In effect, the use of the corrected sea ice height, $hc$, to derive the height of level ice, $hL$, (as described by Duncan and Farrell, 2022) may be counterproductive for the purpose of estimating sail height of grounded ridges.

We addressed tidal range because of the possibility of having consistently biased height measurements in a region if ICESat-2 overflight timing lined up with a particularly high or low tide. We have added discussion of wind-driven changes in local sea level and the impact that would have on the estimates to the manuscript in the Bathymetry section of the Methods (section 2.3), and to our Limitations section (4.5).

A wind-driven surge in local sea level that drives the level ice up around a grounded ridge would likely result in this algorithm missing that ridge. With wind-driven variations in sea level height fairly random in time, we do not expect that would cause a systematic bias in under-detecting ridge heights in a particular area. With ground tracks spread across the Alaska Arctic each day for the five month period of this study, there are likely cases where both sea level is elevated compared to a grounded ridge and where it is depressed compared to a grounded ridge.

The idea of correcting sail heights for dynamic topography of the ocean is an interesting one, and one we would like to pursue further in a future study. From the relative surface height measurements alone, it is unclear which are grounded ridges where the surrounding level ice is elevated versus smaller ridges which do not extend deep enough to be grounded. For tracks where we have reliable retrievals of surface height over shore though, it would be possible to account for the dynamic height and more systematically study the impact of short-term sea level variation on ridge grounding. This would be outside the scope of this paper, but certainly an interesting idea to pursue.

5. GEBCO data are of questionable reliability in shallow water
Given the sparsity of sounding points in shallow water and the year-to-year variability of the bathymetry in these regions, I am somewhat skeptical of the validity of using GEBCO bathymetric data all the way up to the coastline. The GEBCO Type Identifier (TID) Grid, available with GEBCO bathymetry data, show that broad areas of the Alaska coast shallower than ~5 m are coded with the value 41 which indicates the bathymetry values are "Interpolated based on a computer algorithm". I would therefore recommend the authors exercise more caution when identifying grounded features in shallow water. The authors already exclude features in water shallower than the level ice drafter, but I don't understand the rationale for this (see comment 6). Instead, I recommend the authors establish a shallow-water cut-off value, based on the GEBCO TID Grid. This value might vary regionally according to the density of sounding points used in the GEBCO grid, but this would provide a data-driven rationale for excluding ridges in shallow water. Also, under the topic of bathymetric uncertainty, I don't understand how the orange areas in Figures 4-7 were derived. Line 209 states "The orange shading represents positional uncertainty in the bathymetry line", but the width of the orange shading would suggest that the positional uncertainty is on the order of hundreds of meters, which seems too high.

In the 2024 GEBCO product, the TID Grid does not identify shallow water areas as consistently interpolated (Type 41). We have included in this response a map of type identifiers both across northern Alaska and in the area around Utqiagvik and Elson Lagoon (following page). There are singlebeam depth measurements (Type 11) in shallow water ($\leq 2$ m) including in lagoon areas. GEBCO does not establish a shallow-water cutoff. In the revised paper, we note individual ridge detections in figures 5-8 as being either based on singlebeam depth measurements (* markers) or interpolated bathymetry (square markers).

That said, there certainly are issues with depth estimates at shallow water depths. In re-running the analysis we implement a general shallow water cutoff at 3m depth, which is grayed out in figures 6-8 and accounted for in the statistics presented in Table 1, figures 9-11, and the discussion in the results section.. This is in part due to the uncertainty in shallow water measurements, but generally to exclude features inside the lagoon areas. The orange bars in the original version of figures 4-7 reflected the 500m spatial resolution of the GEBCO product. We have removed these and instead address issues regarding the coarse spatial resolution of the GEBCO product in the discussion.

**GEBCO Type ID Map**

[Figure]

0 Land
11 Singlebeam - depth value collected by a single beam echo-sounder
41 Interpolated based on a computer algorithm - depth value is an interpolated value
    based on a computer algorithm (e.g. Generic Mapping Tools)
71 Unknown source - depth value from an unknown source

6. Exclusion of ridges in water shallower than draft of undeformed ice seems unnecessary
I don't understand the rationale for excluding ridges in water shallower than the draft of undeformed ice.
For such ridges to exist at the location, they must have become grounded at some stage when the ice was
thinner. Hence, excluding ridges based in undeformed ice draft creates the possibility of a scenario in
which a ridge is counted as grounded on one day, but excluded the next. I would therefore discourage the
authors from excluding features on this basis and instead simply apply a shallow water cut-off as
explained in comment 5.

We implement a shallow water cutoff of 3m to distinguish grounded ridges from bottom-fast ice along the
coastline. Without a shallow water cutoff in the GEBCO product to use, we found that 3m excluded
features inside lagoon areas and rubble on beaches without excluding too many grounded ridges. Using

5m for the shallow water cutoff excluded a lot more high-likelihood grounded ridges and seemed too high an arbitrary threshold given the data available. This change meant we re-ran the analysis, changing results and discussion throughout the text.

7. Some more careful reading of Mahoney et al (2007a, as cited in the text) may be required
The text makes many references to the work of Mahoney et al (2007a), which I feel mischaracterize what is written in the cited work. For example, on line 308, the text states "Mahoney et al. (2007a) assumes a sail width of 100 m for grounded ridges", but this is a rather inaccurate description of what was written in that paper. For the purposes of estimating the possible spatial density of grounded ridges, Mahoney et al assumed that the keels of ridges deep enough to become grounded were on the order of 100m wide. This is not the same as assuming grounded ridge sails are 100 m wide.

We thank the reviewer for clarification on Mahoney et al.'s results. We revised the reference to Mahoney et al. (2007a) from this paragraph so as to not misrepresent that work (starting line 296) and have added discussion to clarify why our results would differ from this estimate.

Similarly, line 315 begins with the statement "Mahoney et al. (2007a) suggested grounded ridges discontinuously pin the landfast ice edge, roughly every 30 km". In this case, I feel the authors are taking what is written in the cited article somewhat out of context. Instead, Mahoney et al. wrote that grounded ridges would be spaced approximately every 30 km if the thickness distribution of landfast ice was the same as that measured offshore in the drifting pack ice. And they go on to note that grounded ridges are observed more closely spaced than this, suggesting that they are produced through "in-situ" grounding process (i.e. like those illustrated in Figure 1 of this manuscript).

Thank you for this clarification. We have revised the discussion around the frequency of grounded ridges to focus on what we observe in the ICESat-2 tracks (starting at line 380), with the comparison to the ~30km estimate referenced more carefully later in that paragraph.

Lastly, on line 320, the text claims "Mahoney et al. (2007a) assumes that these grounded ridge features are located close to the SLIE", but I again feel this does not accurately represent what Mahoneyet al. wrote. Rather than making any general assumption that grounded ridges are located close to the SLIE, they identify "nodes" where the SLIE occurs most commonly and suggest that these correspond to the location of interannually recurring grounded features. Additionally, I recommend the authors consider citing Mahoney et al's more recent 2014 paper, which expands on the discussion of these nodes and provides photographic documentation of a grounded ridge in the location of a node that happens to be very close to the location of line C in Figure 2. I also feel that some further discussion of the location of the nodes identified by Mahoney et al in relation to the distribution of grounded features shown in Figure 10 would be a valuable addition to the text.

We have added further reference to Mahoney et al.,'s 2014 discussion of nodes in the landfast ice in the discussion. Thank you for sharing the shapefile for the locations of nodes identified in Mahoney et al. 2014 – we have added these to the updated version of Figure 10 (now Figure 11). This figure has been further revised to include three subsets with zoomed-in views of areas around three of the nodes (near longitudes -162, -155, and - 148). The text will be revised to include this additional discussion starting on line 446.

**Minor comments**

Line 9 (and throughout the text): The word "Alaskan" is a noun referring to someone from Alaska. When

used an adjective, the correct term is Alaska.

This is corrected throughout the text.

Figure 1: This figure illustrates one of two ways in which a sea ice ridge can become grounded. The other way involves advection of a deep-keeled ridge into shallow water. Mahoney et al (2007b as cited in the manuscript) speculate that the latter is more likely to create a gouge in the seafloor and is therefore more significant for stabilizing the landfast ice. The authors may wish to acknowledge both ways of creating a grounded ridge and the differences between them.

We acknowledge the advection of deep-keeled ridges in the text, but agree that this can be more explicit and have updated the introduction accordingly. We have added an alternate development track in Figure 1 to make this mechanism more clear, and we have added discussion throughout the manuscript to better reflect this mechanism.

Figure 2: Why did the authors select these ICESat-2 tracks? Had they selected tracks that intersected the coast closer to Utqiagvik, they could fallen within the footprint of UAF's sea ice radar and there's a good chance that they would have intersected whaling trails where sea ice thickness has been routinely measured since 2007 (https://arctic-aok.org/data-sources/whaling-trail-mapping/). These data are both reference could have provided useful validation for the thickness of level ice in the region. Also, why is Line C truncated before the 20m isobath? From several years of observations, the landfast ice in this region is commonly anchored by grounded ridges just beyond the end of Line C.

These tracks were picked from the set in this area that had full repeat coverage showing a variety of sample ridge conditions, as part of the initial case study. Cloud cover/precipitation can prevent surface height retrievals, and these three tracks had very few segments of missing data in the regions around Utqiagvik for ground tracks both in January and April of that year. There are not available tracks with full coverage in both January and April that are closer to the Utqiagvik sea ice radar during the 2021-2022 winter. We do have a pair of tracks from December and March, but the December track is prior to the formation of grounded ridges.

There are a few single tracks (without repeat) that intersect the sea ice radar area. Grounded ridges detected in the sea ice radar area are approximately 1.2km from shore: this is consistent with the persistent features in the sea ice radar. It is harder to see those features in individual images in the sea ice radar (more apparent in the video animation), so we are not going to put an ice radar image in the paper, but we have added discussion to section 4.1 (Detection of Grounded ridge features) describing this additional means of validation.

We have updated the figure for Line C beyond the 20m isobath. In the January track, there are no further grounded ridges beyond what was shown in the prior version. In April, there is one ridge at 14km from shore that barely intersects the bathymetry line for the largest keel:sail ratio estimate ('low' confidence ridge) at 16m bathymetric depth. We have updated results section 4.3 accordingly.

The whaling trail mapping effort has a lot of potential for a larger validation study of this approach: while we were only able to process a single year of the ICESat-2 data for this project, we hope to secure funding for a larger effort covering more years and more area. Working with the local community for detailed sail height measurements (and any keel geometry information) would be an important part of that.

Lines 265-267: There are multiple assertions here that should be supported by references. I think I know what the authors mean by the "classic" ridge, but I have a suspicion that its classic status derives from simplifications adopted in many illustrations of coastal ice over the years together with a bias in the early literature toward ridges in the Beaufort Sea. Also, can the authors provide a reference supporting the "prevalance of shear" on the Chukchi Side of Point Barrow? The climatological prevailing wind is from the east, with creates a lot more divergence than shear and a lot less shear than on the Beaufort side of Pot Barrow

We reviewed the sea ice radar animations for the 2021-2022 winter: while there is very little overlap with the ice radar field of view and the ICESat-2 track (and it is hard to determine exactly where it is without geolocated radar data), the radar does show a persistent feature around the same distance from shore. We've added reference to that starting on line 280.

Line 337-338: The reasoning behind this explanation in the final sentence of this paragraph is not clear to me. If grounded ridges in the same water depth are higher in the Beaufort Sea than in the Chukchi, this means the ice is piled higher above the waterline in the Beaufort. I can envision a few mechanisms that might cause such a difference (for example in-situ grounding vs advection of deep-keeled ridges in shallow water) and how they might relate to coastal aspect or parent ice thickness, but I feel most readers would benefit from further explanation.

We appreciate this feedback and have added additional explanation to the text starting on line 397. The change keel:sail ratios used for determining which ridges are grounded changed the distributions here slightly, which is reflected in the text in the paragraph starting at line 405.

---

## Referee Report (RR1)

**Grounded ridge detection and characterization along the Alaska
Arctic coastline using ICESat-2 surface height retrievals**

by Kennedy A. Lange, Alice C. Bradley, Kyle Duncan, and Sinéad L. Farrell

2nd version
*Re-submitted to The Cryosphere Discussions*
*https://doi.org/10.5194/egusphere-2024-1885*

***Review***
*Jan 9, 2025*

Summary
    The authors have done a commendable job of addressing all the comments in my first review as well as those of my co-reviewer, Dr. Lemieux. I am especially impressed with their approach to assigning confidence levels to their grounded ridge detections, which greatly strengthens their interpretations of the spatial distribution of grounded ridges in their study area. In my opinion, this manuscript is very close to ready for publication, but I have outlined a few additional comments below that I feel should be addressed first. As far as possible, I have tried to include clear recommendations with each comment and I believe the authors will find them all simpler to address than those in my first review.

Major Comments

1. Confusing grounded ridge legend (Figures 5-8)
    I really like the addition of the colored regions to indicate keel depths derived from different sail height ratios on Figures 5-8, but I find the symbology used to assign confidence and bathymetry data type to each grounded ridge somewhat confusing. First, it took me a little while to understand the meaning of the purple and yellow squares in Figure 5, when neither of these symbols is included in the legend. Reading between the lines, I assume the color of the symbol designates the confidence level, while the shape indicates the nature of the bathymetry data. If so, this should be spelled out clearly. However, as an alternative that might require less explanation, I recommend using separate symbols above and below each ridge to provide this information. For example, you could use a single color-coded symbol above each ridge to indicate confidence and perhaps a letter character below the keel for bathymetry data type. Please also see my associated comment 2 below.

    My second source of confusion is the presence of multiple symbols for each ridge. In Figure 5. this appears to indicate both ridges are identified with both low and medium confidence, but the caption states "These ridge features are characterized as medium confidence grounded ridges". I'm not sure why ridges would be assigned more than one confidence level, but unless there's an aspect to this that I'm missing, I would recommend the authors identify only the highest level.

    Lastly, the locations of grounded ridges in the SAR imagery in Figures 6-8 is only marked with purple dots, potentially creating the impression that they are all high confidence features. Instead, you could use the same symbols as those used above each ridge (per my suggestion above) to indicate ridge locations.

2. GEBCO Type Identifier (TID) grid data

I appreciate the authors including an illustration of the GEBCO 2024 TID grid in their response, which shows more shallow water single-beam data than I had known about. Nonetheless, there are still some areas of the coastline within this paper's study region that have few direct observations of bathymetry. In particular, the figure below highlights three areas with very few single beam data and primary reliance on interpolation. I realize this is something of a moot point now that the authors are using a deeper shallow-water cutoff for their analysis, but the figure below also illustrates that there is a greater diversity of data sources in the grounding regions than the 3 types that the authors discuss in their response and in the manuscript. Given this diversity, I recommend adopting GEBCO's categorization scheme and simply distinguishing between grid cells with "direct", "indirect", and "unknown" bathymetry measurements. Per comment 1 above, these could be designated with "d", "i", or "u".

[Figure]

3. Missing details on SAR data products and radiometric processing

This is a somewhat minor point, but since Figures 6-8 provide a color scale bar in dB (see minor comment below), I recommend adding some brief details about the acquisition mode and polarization of the data as well as the processing options applied (e.g., gamma-nought or sigma-nought for backscatter coefficient). The text should also state the source of the Sentinel-1 SAR data (i.e., ESA or ASF, or elsewhere) with appropriate link or DOI.

**Minor comments**

Figure 1: I like this new figure, but I have two minor comments. First, I feel the reader might benefit from some additional annotations to indicate which ice is mobile and which ice is stationary in each of the sub-diagrams A-D. This could help clarify that the ridge is part of the drifting pack ice in panel B, but is forming at the boundary between the landfast ice and pack ice in panel C. Second, the label for "Shorefast / landfast ice" should extend all the way from the shoreline to the SLIE. In other words, bottom fast ice, grounded ridges, and the shorefast ice extension are all different parts of the shorefast ice.

Figures 6-8: The color scale bar for the SAR imagery indicates a range of 12-22 dB, which I believe is in error. I expect the range should -22 to -12 dB.

Lines 508-510: This statement about differences in the size and number density of ridges in the Chukchi and Beaufort Sea is very interesting, but I feel it requires some additional citation or other support. Is this something the authors found from their own data, or are they describing findings made by others?

Lines 589-591: I agree with your revised description of the impact of sea level variations on sail height and therefore grounded ridge detection. However, I feel the key sentence could use some clarification regarding the direction of sea level change. Here is a suggestion for rewording (strikethrough indicates deleted text and bold indicates new text):

"Assuming the level ice can float independently ,  **positive** variations **in sea level** would  **lift** the level ice **around the grounded ridge, thereby reducing the relative** height **of the** ridge sail  (Hs),  and  **leading to an** underestimat **of** keel depth (Hk)."

---

## Author Response (AR2)

We thank both reviewers for their thoughtful and detailed comments. Your help has really made this a better paper!

Comments are addressed point-by-point below, with our responses highlighted for easier reading.

KL, AB, SF, KD

JL: MINOR COMMENTS

1) Sometimes you write landfast and sometimes land-fast. Please make these consistent.

We have corrected this to be 'landfast' throughout the paper.

2) Fig.6-8: I understand that the stars and squares identify the level of confidence of grounded ridges. How don't understand why, for example, a grounded ridge would have stars of different colors. If it the confidence is high then I would just show the star for the high confidence.

We have updated figures 5-8 consistent with both reviewers' recommendations: the ridge identifiers (Δ) only show the highest-confidence classification for a ridge, and the bathymetry type designator is now in a row at the bottom of the figure.

3) Fig. 6: The bathymetry is different between the two tracks (like from 2.5 km to 3.5 km).Is it because there is no perfect overlap between the tracks? It would be nice to clarify this for Fig. 6.

The two tracks are close to co-located, but there is a small amount of distance between them.
(We have spent a lot of time verifying that these differences in bathymetry are the result of slight differences in the ground tracks between the two passes). We have added mention of this to the text in line 279.

4) lines 306-307 and lines 309: I have the impression you repeat the same information.

We rephrased those lines to reduce repetition (now starting in line 314).

5) line 356: add a space after 6.17 km.

Thanks for catching this (line 361).

6) Fig. 9: why are some dots not connected with the lines?

There are some bins in the histogram that have no observations: dots not connected by lines indicate gaps in the observed distribution.

7) Fig. 9: These are PDF. They don't show directly the probability but the probability density. I would write PDF for the y-labels and write 'probability density function (PDF)' in the caption.

Thank you for catching this error: it has been corrected in Figure 9 and the caption.

8) Fig. 10: the y-labels 'Surface Height' should be replaced by 'Sail Height'.

This has been corrected in Figure 10 and the caption.

9) lines 416-417: rephrase.

This has been rephrased to "A similar seasonal trend is present in Beaufort sail heights, increasing from 1.5 m in December to 2.2 m in April." on line 420/421

10) line 421: replace 'ridge height' by 'sail height'.

This has been corrected in line 426

11) Fig. 10 and text associated with it: I think this figure is very interesting but requires some clarifications. This is only for grounded ridges, right? This is what the caption says but the titles above the columns are confusing. On line 410, it is written 'seasonal trend in ridge height, depth,...'. I would write 'seasonal trend in grounded ridge...'.

We have retitled Figure 10's columns "Grounded Ridges" and "Grounded Ridge Furthest From Shoreline" to clarify which data is plotted. We have rephrased line 410 to read as "Next, we show that seasonal trends in grounded ridge sail height, depth, distance from shore, and width can be delineated from the data with infrequent repeat measurements over the same geographic area." This is now line 415.

12) lines 412-413 (related to Fig. 10): it is not clear to me that depth (and maybe distance) tend to increase during the season. Please modify text if needed. The text on lines 419-420 contradicts this.

Thank you for pointing this out, we have added some additional text to clarify our results. We write "The mean distance from shore and width calculated based on all grounded ridges is variable throughout the season, but there are more evident trends when considering only the grounded feature furthest from shore along each track." We hope that this will shed light on the fact that the increasing trend is more present when accounting only for ridges furthest from shore. This is in line 424/425 now.

13) line 421: replace 'ridge height' by 'sail height' and add 'grounded' before 'ridge furthest seaward'...

This has been corrected in the text.

14) line 434: add 'during' after 'point'.

This has been corrected in the text.

15) the reference Beatty and Holland (2010) should be König Beatty and Holland (2010).

Thank you, this has been corrected in the text.

16) Fig. 11: the y-label 'Height' should be replaced by 'Sail Height'.

This has been updated on Figure 11.

AM comments:

**Major Comments**

1. Confusing grounded ridge legend (Figures 5-8)
I really like the addition of the colored regions to indicate keel depths derived from different sail

height ratios on Figures 5-8, but I find the symbology used to assign confidence and bathymetry data type to each grounded ridge somewhat confusing. First, it took me a little while to understand the meaning of the purple and yellow squares in Figure 5, when neither of these symbols is included in the legend. Reading between the lines, I assume the color of the symbol designates the confidence level, while the shape indicates the nature of the bathymetry data. If so, this should be spelled out clearly. However, as an alternative that might require less

explanation, I recommend using separate symbols above and below each ridge to provide this information. For example, you could use a single color-coded symbol above each ridge to indicate confidence and perhaps a letter character below the keel for bathymetry data type. Please also see my associated comment 2 below.

My second source of confusion is the presence of multiple symbols for each ridge. In Figure 5. this appears to indicate both ridges are identified with both low and medium confidence, but the caption states "These ridge features are characterized as medium confidence grounded ridges". I'm not sure why ridges would be assigned more than one confidence level, but unless there's an aspect to this that I'm missing, I would recommend the authors identify only the highest level.

Lastly, the locations of grounded ridges in the SAR imagery in Figures 6-8 is only marked with purple dots, potentially creating the impression that they are all high confidence features. Instead, you could use the same symbols as those used above each ridge (per my suggestion above) to indicate ridge locations.

We have updated figures 5-8 consistent with your suggestions: the ridge identifiers (Δ) only show the highest-confidence classification for a ridge, and the bathymetry designator is now in a row at the bottom of the figure. Rather than using letters (the font size necessary to avoid overlap was very small), we used symbols of | for a direct measurement, – for an indirect measurement, and ? for an unknown measurement.

2. GEBCO Type Identifier (TID) grid data
I appreciate the authors including an illustration of the GEBCO 2024 TID grid in their response, which

shows more shallow water single-beam data than I had known about. Nonetheless, there are still some areas of the coastline within this paper's study region that have few direct observations of bathymetry. In particular, the figure below highlights three areas with very few single beam data and primary reliance on interpolation. I realize this is something of a moot point now that the authors are using a deeper shallow-water cutoff for their analysis, but the figure below also illustrates that there is a greater diversity of data sources in the grounding regions than the 3 types that the authors discuss in their response and in the manuscript. Given this diversity, I recommend adopting GEBCO's categorization scheme and simply distinguishing between grid cells with "direct", "indirect", and "unknown" bathymetry measurements. Per comment 1 above, these could be designated with "d", "i", or "u".

We had grouped together the interpolated categories, but since the vast majority of direct observations in the region were single-beam, we missed that there were a few places with other direct measurements. We have fixed the figures and discussion to call them direct measurements rather than single-beam measurements per your suggestion. Rather than using letters (the font size necessary to avoid overlap was very small), we used symbols of | for a direct measurement, – for an indirect measurement, and ? for an unknown measurement.

3. Missing details on SAR data products and radiometric processing

This is a somewhat minor point, but since Figures 6-8 provide a color scale bar in dB (see minor comment below), I recommend adding some brief details about the acquisition mode and polarization of the data as well as the processing options applied (e.g., gamma-nought or sigma- nought for backscatter coefficient). The text should also state the source of the Sentinel-1 SAR data (i.e., ESA or ASF, or elsewhere) with appropriate link or DOI.

We added a brief description of the acquisition mode to the paragraph describing the SAR imagery starting at line 129. We have cited the Sentinel-1 SAR data consistent with the ASF's recommended citation (this was in the file previously, but didn't play nice with the LaTeX template. It's fixed now – reference to ESA 2022) and have added a note in the text that we accessed the data through ASF.

We also removed the numerical label on the dB color scale in figures 6-8: see further note following the minor comment below.

**Minor comments**

Figure 1: I like this new figure, but I have two minor comments. First, I feel the reader might benefit from some additional annotations to indicate which ice is mobile and which ice is stationary in each of the sub-diagrams A-D. This could help clarify that the ridge is part of the drifting pack ice in panel B, but is forming at the boundary between the landfast ice and pack ice in panel C. Second, the label for "Shorefast / landfast ice" should extend all the way from the shoreline to the SLIE. In other words, bottom fast ice, grounded ridges, and the shorefast ice extension are all different parts of the shorefast ice.

We have color-coded the ice in this diagram (Figure 1) to be light gray for stationary and white for drifting, in order to make it clear throughout the diagram

what is drifting versus shorefast ice. We have also fixed the shorefast/landfast ice label.

Figures 6-8: The color scale bar for the SAR imagery indicates a range of 12-22 dB, which I believe is in error. I expect the range should -22 to -12 dB.

We removed the numeric labels on the SAR images: the use here is to use pattern matching to monitor the persistence of features in the shorefast ice throughout the season in order to see when features form and break up. The imagery shown is 12-22 dB above background noise levels. Rather than get distracted by the processing details when they weren't particularly relevant to the paper, we omitted the numerical scale bar.

Lines 508-510: This statement about differences in the size and number density of ridges in the Chukchi and Beaufort Sea is very interesting, but I feel it requires some additional citation or other support. Is this something the authors found from their own data, or are they describing findings made by others?

This was found both in our data and is consistent with Strub-Klein and Sudom 2012. We added a note in that paragraph (which is referring to the data shown in figure 9) to that effect, on line 409.

Lines 589-591: I agree with your revised description of the impact of sea level variations on sail height and therefore grounded ridge detection. However, I feel the key sentence could use some clarification regarding the direction of sea level change. Here is a suggestion for rewording (strikethrough indicates deleted text and bold indicates new text):

"Assuming the level ice can float independently, positive variations **in sea level** would **lift** the level ice **around the grounded ridge, thereby reducing the relative** height **of the** ridge sail (Hs), and **leading to an** underestimat**e of** keel depth (Hk)."

We have corrected this in the text at line 480